# GAPS: FEW-SHOT INCREMENTAL SEMANTIC SEGMENTATION VIA GUIDED COPY-PASTE SYNTHESIS

## ABSTRACT

Few-shot incremental segmentation is the task of updating a segmentation model, as novel classes are introduced online overtime with a small number of training images. Although incremental segmentation methods exist in the literature, they tend to fall short in the few-shot regime and when given partially-annotated training images, where only the novel class is segmented. This paper proposes a data synthesizer, Guided copy-And-Paste Synthesis (GAPS), that improves the performance of few-shot incremental segmentation in a *model-agnostic* fashion. Despite the great success of copy-paste synthesis in the conventional *offline* visual recognition, we demonstrate substantially degraded performance of its naïve extension in our *online* scenario, due to newly encountered challenges. To this end, GAPS (i) addresses the partial-annotation problem by leveraging copy-paste to generate fully-labeled data for training, (ii) helps augment the few images of novel objects by introducing a guided sampling process, and (iii) mitigates catastrophic forgetting by employing a diverse memory-replay buffer. Compared to existing state-of-the-art methods, GAPS dramatically boosts the novel IoU of baseline methods on established few-shot incremental segmentation benchmarks by up to 80%. More notably, GAPS maintains good performance in even more impoverished annotation settings, where only single instances of novel objects are annotated.

## 1 INTRODUCTION

Incremental segmentation is an important capability for open-world AI systems. For example, consider a housekeeping robot that has been trained to segment common household objects, but once deployed in a user's home it encounters a previously unseen type of furniture. For such practical applications, incremental segmentation would be capable of expanding the set of recognized classes to contain the new object. There are a few desired properties of incremental segmentation algorithms to operate under these scenarios. First of all, the algorithm should be equipped with **few-shot learning capability**, which means that the algorithm can benefit from as few as one image provided by a user rather than requiring hundreds of images annotated offline by professional annotators. Second, providing full segmentation annotation of an image is time-consuming. To avoid causing substantial burdens for untrained users, the algorithm needs to be trainable with **partially-annotated** images where only novel classes are annotated.

A few attempts have been made by recent works (Cermelli et al., 2020; Cha et al., 2021; Douillard et al., 2021; Zhang et al., 2022; Yan et al., 2021) on *non*-few-shot incremental segmentation to investigate learning with partially-annotated images, which is termed *semantic background shift* (Cermelli et al., 2020). Background shift describes a challenge unique to incremental semantic segmentation where classes that are not in the current learning step are assigned 'background' labels, which prohibits direct end-to-end training. Recent work uses either modified loss (Cermelli et al., 2020; Zhang et al., 2022) or pseudo-labeling (Cha et al., 2021; Douillard et al., 2021; Yan et al., 2021) as *proxies* to train on partially-annotated images. However, although these proxying methods demonstrate good performance under the non-few-shot settings, they rely on rich annotations and fall short when only a limited amount of data is presented to the model, due to a lack of diversity of data. An even more restrictive setting occurs when users label only a single instance of the novel class, which can dramatically hurt performance of proxy models, due to the training containing non-annotated instances of the novel class (which are treated as negative pixels).

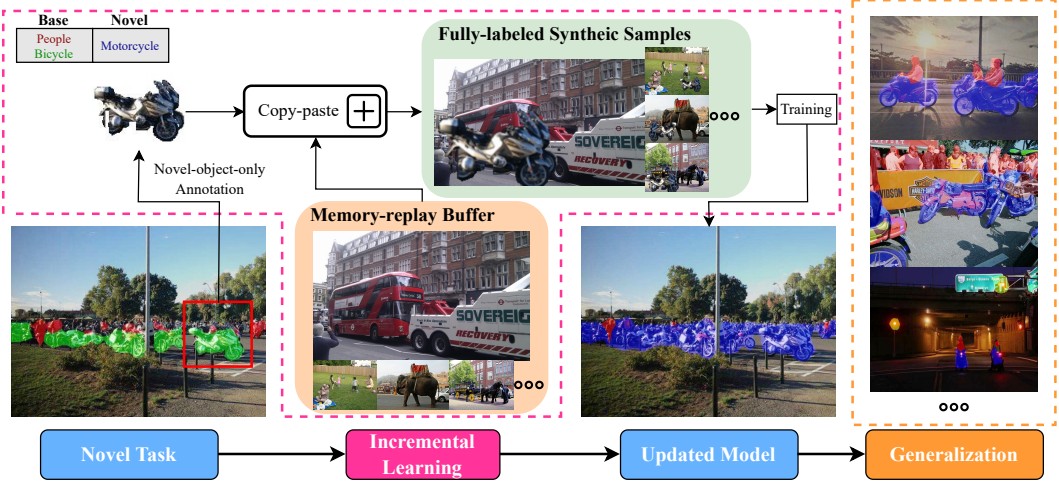

Figure 1: Our proposed method utilizes guided copy-paste augmentation to synthesize diverse training data, using as few as **one single novel instance** for training. For example, the model encounters an image of many motorcycles, which is novel to the model. As a result, the model incorrectly assigns learned bicycle labels to these pixels and therefore needs to be updated. Our proposed method can adapt to the novel motorcycle class with an annotation of a single motorcycle, which can be efficiently annotated; whereas previous work (Cermelli et al., 2020; 2021) require time-consuming annotation of all instances of motorcycles or even the entire image. Best view in color.

To address the aforementioned challenges, we propose GAPS (Guided copy-And-Paste Synthesis), which improves the training of incremental segmentation models by synthesizing fully-annotated images from partially-annotated examples. It is *model-agnostic*, and can be inserted as a plug-and-play module into different incremental learning algorithms, e.g., standard fine-tuning or PIFS (Cermelli et al., 2021). Copy-paste generates diverse training data to boost performance under few-shot settings, enables the model to learn with partially-annotated images with as few as one annotated novel instance out of many novel instances in an image (e.g., as illustrated at the lower left part of Fig. 1), which is a *stricter* setting than semantic background shift (Cermelli et al., 2020).

To the best of our knowledge, we are the *first* to introduce copy-paste as a synthesis technique to create a diverse data source for few-shot incremental segmentation. Although copy-paste (Ghiasi et al., 2021) has been shown to be an effective data augmentation technique for offline visual recognition tasks, we identify new key technical challenges to adapting it to few-shot incremental settings. First, how should the synthesizer pick representative samples from the base dataset to construct a *diverse* pool of fully-annotated base scenes? Second, given the constructed pool of fully-annotated images, how should it select the most *suitable* base images to be pasted on? Third, after an informative image is selected, from what distribution should it sample current and previously learned novel objects to *balance* sample frequency and avoid over-sampling or under-sampling? Our GAPS method differs from a naïve (e.g., uniform random sampling) copy-paste process by a *guided* strategy that considers diversity of the memory-replay buffer, imbalanced class frequencies between base classes and novel classes, and contextual similarity of images.

In summary, our contributions are as follow:

1. We are the first to introduce copy-paste as a synthesis technique to address partially-labeled images for incremental segmentation.

2. To address the gaps between copy-paste under the offline setting as an augmentation technique and under the online setting as a synthesis technique, we design a guided copy-paste process that improves the distribution of synthesized images by enforcing diversity of the memory-replay buffer, exploiting contextual information, and balancing class frequencies.

3. The proposed GAPS technique consistently boosts the performance of a variety of incremental learning algorithms from simple fine-tuning to sophisticated state-of-the-arts under

the few-shot setting. Furthermore, we demonstrate the strength of GAPS to cope with a more challenging task setting where only one instance out of many novel instances in an image is annotated, which highlights copy-paste as a better alternative to pseudo-labeling or modified loss for practical incremental segmentation applications.

## 2 RELATED WORK

**Incremental Learning for Semantic Segmentation.** It is known that many learning-based models suffer from catastrophic forgetting (McCloskey & Cohen, 1989), a phoenomenon that causes models to perform significantly worse on old tasks when they are fine-tuned to adapt to new tasks. *Incremental learning* studies how to enable models to adapt to new classes while mitigating catastrophic forgetting without accessing the old dataset or full-scale re-training. This problem has been studied extensively in image classification (Li & Hoiem, 2017; Lopez-Paz & Ranzato, 2017; Yoon et al., 2018; Rebuffi et al., 2017; Mallya et al., 2018; Castro et al., 2018; Bang et al., 2021); whilst relatively less work have been done to study incremental learning under the task setting of semantic segmentation (Cermelli et al., 2020; Michieli & Zanuttigh, 2019; Cha et al., 2021; Douillard et al., 2021; Zhang et al., 2022). Noticeably, a few attempts have been made by recent work to address the semantic background shift problem proposed by Cermelli et al. (2020) via either pseudo-labeling (Cha et al., 2021; Douillard et al., 2021; Yan et al., 2021) or modified loss (Cermelli et al., 2020; Zhang et al., 2022) to train on partially-annotated images of novel classes. However, existing work relies on rich annotations and tends to fail when only a limited amount of data is available. In contrast, our work enables incremental segmentation learning with *few data* via a guided copy-paste process, which demonstrates promising performance under the few-shot and more impoverished single-instance setting. Furthermore, GAPS is a *model-agnostic data pre-processor*, which is orthogonal to incremental learning techniques such as regularization (Li & Hoiem, 2017).

**Few-Shot Semantic Segmentation.** *Few-shot semantic segmentation* methods predict segmentation masks of novel classes using only a few training examples of the novel class. Many meta-learning-based methods (Shaban et al., 2017; Wang et al., 2019; Tian et al., 2020b; Zhang et al., 2020) and even specialized datasets (Li et al., 2020) have been proposed to address such a problem. However, few-shot semantic segmentation methods produce novel-class-only binary foreground-background segmentation. In comparison, our proposed method works in a more challenging and realistic setting where both base classes and novel classes need to be segmented.

**Few-Shot Incremental Segmentation.** While there are many works in few-shot incremental image classification (Tao et al., 2020; Cheraghian et al., 2021), relatively fewer works have been done to investigate few-shot incremental segmentation (Tian et al., 2020a; Cermelli et al., 2021; Ganea et al., 2021). Tian et al. (2020a) designs a meta-learning-based classifier that adjusts learned prototypes by modeling interaction between base classes and incoming novel class. Unlike Tian et al. (2020a), which only performs a single update of weights in the classifier, PIFS (Cermelli et al., 2021) apply regularization techniques to allow fine-tuning of the entire network, achieving state-of-the-art result in few-shot incremental semantic segmentation. However, PIFS (Cermelli et al., 2021) is fine-tuned on only a small number of samples, which leads to sub-optimal performance due to overfitting. In addition, PIFS requires fully-annotated images as input, which hinders its potential for practical applications.

**Copy-Paste Augmentation.** Copy-and-paste is an augmentation technique that copies a subset of objects from one image and pastes onto the other image using their segmentation masks. Many works (Dwibedi et al., 2017; Ghiasi et al., 2021; Dvornik et al., 2018) have been done to investigate how copy-and-paste augmentation can help with various visual tasks. Dvornik *et al.* apply copy-paste augmentation in object detection by designing a neural network to consider context and guide copy-paste. However, the context guidance method proposed by Dvornik *et al.* can not be trivially applied to our application since it requires abundant fully-annotated training data. More recently, Ghiasi *et al.* conduct extensive experiments to demonstrate the effectiveness of simple copy-paste in the instance segmentation problem. We extend the augmentation strategy from Ghiasi et al. (2021) and construct an intuitive baseline called Naïve copy-Paste Synthesis (NPS) to adapt it to our online task setting. However, as we will demonstrate, such naïve adaptation gives unsatisfactory performance in our task setting because of *gaps* between the static offline learning and continual

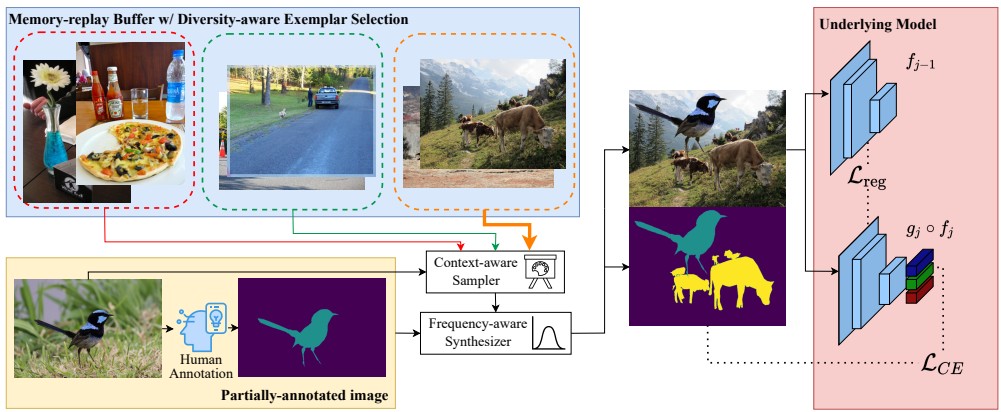

Figure 2: Overview of GAPS. During the incremental learning stage, GAPS takes in as few as one annotated instance of a single image. It is more probable for GAPS to select a scene contextually similar to the provided image from memory-replay buffer $\hat{D}_0$. The image is then probabilistically pasted to generate synthetic fully-labeled scenes. Note that GAPS is model-agnostic, and here we use PIFS (Cermelli et al., 2021) as an example for the underlying segmentation model to illustrate how GAPS is applied as a pre-processor. Best seen in color.

online learning. In our work, we propose a series of techniques to guide the copy-paste synthesizer to address these gaps, whose effectiveness is evident from the significant improvement from NPS.

## 3 METHOD

**Problem Setup.** Let $\mathcal{X} \subset \mathbb{R}^{H \times W \times 3}$ be a set of RGB images with size $H \times W$, $\mathcal{C} \subset \mathbb{N}$ be a set of category labels, and $\mathcal{Y}^{\mathcal{C}} \subset \mathbb{R}^{H \times W \times |\mathcal{C}|}$ be a set of label masks (*i.e.*, per-pixel category labels in $\mathcal{C}$). In semantic segmentation, we aim to learn a model $\phi$ that maps an image $x \in \mathcal{X}$ to a segmentation mask $y \in \mathcal{Y}^{\mathcal{C}}$. Different from standard semantic segmentation, in few-shot incremental segmentation, $\mathcal{C}$ is expanded over time through two stages. During the *base learning stage*, the model is provided with a base dataset $\mathcal{D}_0 = \{(x_i, y_i) | x_i \in \mathcal{X}, y_i \in \mathcal{Y}^{\mathcal{C}_0}\}$, where $\mathcal{C}_0$ is a set of classes in the base dataset. $\mathcal{D}_0$ generally contains many fully-annotated image-mask pairs and is used to train the model $\phi_0 : \mathcal{X} \to \mathcal{Y}^{\mathcal{C}_0}$ from scratch.

During the *incremental learning stage*, a sequence of tasks $\{D_1, D_2, \dots\}$ with novel categories is presented to the model, where $D_j = \{(x_i, y_i) | x_i \in \mathcal{X}, y_i \in \mathcal{Y}^{\mathcal{C}_j}\}$ and $\mathcal{C}_j$ is a set of classes for task $D_j$. In few-shot learning, the size of the training sets for the novel tasks is small, *i.e.*, $|D_j| \ll |\mathcal{D}_0|$. After adapting to task $D_j$, the model is updated as $\phi_j : \mathcal{X} \to \mathcal{Y}^{\cup_{i=0,\dots,j} \mathcal{C}_i}$. The goal of incremental learning is to optimize the model performance jointly on both previous tasks and the current task. To enforce the partially-annotated image setting, we follow Cermelli *et al.* and assume that only novel classes are annotated, *i.e.*, $\mathcal{C}_i \cap \mathcal{C}_j = \varnothing$ for all $i \neq j$.

**Method overview.** Fig. 2 illustrates our proposed Guided copy-Paste Synthesis (GAPS) framework for few-shot incremental segmentation. It is a *generic and model-agnostic* data synthesis framework that generates fully-labeled scenes from partially-annotated images of novel objects as a preprocessor to the underlying segmentation model. After the standard base learning stage with base dataset $D_0$, we build a memory-replay buffer $\hat{D}_0$ using an *diversity-guided exemplar selection strategy* (Section 3.2). During the incremental learning stage, fully-labeled samples are synthesized by copying from the masked novel objects in $D_1, \dots, D_j$ and pasting onto base exemplars from the replay buffer $\hat{D}_0$. The strategy by which we choose base exemplars and novel segments is *context-guided* (Section 3.3) and *class-frequency-guided* (Section 3.4).

## 3.1 Few-Shot Incremental Segmentation Model

In principle, GAPS is model-agnostic, which means that it can work with many incremental segmentation models as a diverse data source to improve their performance. Here we adopt PIFS (Cermelli et al., 2021) as the main baseline underlying segmentation model for its state-of-the-art performance on few-shot incremental semantic segmentation and support for end-to-end training. The PIFS segmentation model $\phi$ is composed of a convolution-based feature extractor $f$ and a per-pixel classification layer $g$ using prototypical representation – $g$ is configured to classify the pixels into $n$ classes, so it is parameterized with prototypes $W = [w_1, w_2, \ldots, w_n]$. Intuitively, $f$ maps every pixel in an input image onto the unit hyper-sphere in a high-dimensional representation space. $g$ then generates probability prediction by comparing cosine similarity of feature vectors with learned class prototypes $w_i$ in the representation space and applying softmax of the resulting similarities.

We want to note that our re-implementation of Cermelli et al. (2021) uses $\mathcal{L}_2$ regularization rather than the prototype distillation loss proposed by Cermelli *et al*. We found experimentally that when a diverse data source is used (i.e., our proposed GAPS), $\mathcal{L}_2$ regularization works better. To be more precise, we construct a penalization term $\mathcal{L}_{\text{REG}}$ to regularize the output before the classifier. For incremental learning task $D_j$ with image-mask pairs $(x, y)$, we have

$$\mathcal{L}_{\text{REG}} = ||f_j(x) - f_{j-1}(x)||_2. \tag{1}$$

The final training loss is given by

$$\mathcal{L}(x, y) = \mathcal{L}_{\text{CE}}(\phi_j(x), y) + \lambda \mathcal{L}_{\text{REG}}, \tag{2}$$

where $\mathcal{L}_{\text{CE}}$ is either the standard cross-entropy loss or the modified cross-entropy loss from Cermelli et al. (2020). $\lambda$ is a hyper-parameter used to weight the regularization loss. All other components are the same as in (Cermelli et al., 2021). We denote our re-implementation of PIFS with $\mathcal{L}_2$ regularization loss as PIFS($\mathcal{L}_2$).

## 3.2 Diversity-guided Exemplar Selection with Learned Prototypes

For methods with memory-replaying (e.g., SSUL (Cha et al., 2021)), GAPS can work directly on top of their constructed buffers with minimal modification. For other methods such as PIFS (Cermelli et al., 2021), we propose a diversity-guided exemplar selection process that builds a small yet diverse memory-replay buffer $\hat{D}_0$ from $\mathcal{D}_0$ to mitigate catastrophic forgetting. Selecting diverse examples that are representative of the base dataset helps mitigate catastrophic forgetting, as suggested by Rebuffi et al. (2017). Inspired by Bang et al. (2021), we select samples distributed uniformly along a spectrum from easy to hard for diversity.

Here, we present an algorithm (Algorithm 1) to construct $\hat{D}_0$ by exploiting the Masked Average Pooling (MAP) function from Cermelli et al. (2021). Intuitively, we approximate the difficulty of every sample by their similarity between the estimated prototype with learned prototypes. Estimated prototypes that are close to the learned prototype are considered easy samples and vice versa. After building a list of base samples sorted by difficulties, we select samples from equally-spaced intervals to ensure samples of all difficulties are selected for diversity.

---

**Algorithm 1** Construct Memory-replay Buffer

**Require:** number of exemplars $n$
  $k \leftarrow \text{FLOOR}(n/|Y_0|)$ // Sample per class
  **for** $c$ from 1 to $|Y_0|$ **do**
      $S_c \leftarrow \{(x_i, y_i) \in D_0, c \in y_i\}$
      **for** $(x_i, y_i) \in S_c$ **do**
          $p_i \leftarrow \text{MAP}(x_i, y_i, c)$ // Pred. Proto.
          $s_i \leftarrow \text{COSINESIMILARITY}(p_{ic}, w_c)$
      **end for**
      Sort $S_c$ by similarity score $s_i$
      $ES_c \leftarrow \{\}$ // final exemplar set of class $c$
      **for** $j = 1, 2, \ldots, k$ **do**
          $L_{idx} \leftarrow j \cdot |S_c|/k$
          $U_{idx} \leftarrow \text{MIN}(Lower + |S_c|/k, |S_c|)$
          $(x, y) \leftarrow \text{SAMPLE}(S_c[L_{idx} : U_{idx}])$
          $ES_c \leftarrow ES_c \cup (x, y)$
      **end for**
  **end for**
  $\hat{D}_0 \leftarrow \text{UNIFORMSAMPLE}(\bigcup_{i=1,\ldots,|Y_0|} ES_i, n)$

---

During the incremental learning stage, we select at most $k$ samples for each novel class using the same algorithm to memorize novel classes. To maintain the size of the memory-replay buffer, we remove old samples from the memory-replay buffer but keep at least $80\%$ of the samples to be fully-annotated samples, so that we have diverse base images for copy-pasting.

### 3.3 Context-guided Sampling

We hypothesize that synthesizing novel objects onto contextually consistent base images would result in an improved learning process. For example, a TV in an apartment should more likely be pasted onto an image of another apartment rather than an outdoor landscape. We design a context-guided sampling algorithm to select images from $\hat{D}_0$ that are contextually similar to the provided partially-labeled images.

GAPS uses a scene embedding network $h : \mathcal{X} \to \mathbb{R}^m$ to estimate pairwise contextual similarity between any two given images. The embedding $h$ maps an input image to a scene embedding vector in a metric space where pairwise similarity comparison between two embeddings are possible. In our system, we use an off-the-shelf VGGNet (Simonyan & Zisserman, 2015) and replace its last fully connected layer with a cosine-similarity-based classifier. The network $h$ is trained using the Places365 (Zhou et al., 2017a) dataset, which contains 365 scene categories and roughly 1.6 million images in its training split. During training, the scene prediction is done by comparing the predicted embedding with existing prototypes. After training, the network $h$ is frozen and the 365 prototypes are discarded.

To find contextually similar base images to each novel image, we evaluate the cosine similarity of the novel image to each of the examples in $\hat{D}_0$, and constuct a contextually similar subset $\mathbb{S}$ with $|\hat{D}_0|/10$ most contextually similar examples. When there are multiple novel images, we take a union of selected examples. To allow other base scenes to be sampled to mitigate catastrophic forgetting, we sample from $\mathbb{S}$ with a probability of $\alpha$, and sample from $\hat{D}_0$ with a probability of $1 - \alpha$, where $\alpha$ is a hyperparameter set to $0.9$ in our implementation. Note that we only need to compute scene embedding once for every image in $\hat{D}_0$ and incoming partially-annotated images. Hence, the context-guided sampling algorithm poses only minor computational overhead to GAPS.

### 3.4 Class-frequency-guided Probabilistic Synthesis

Now the final question is, given a fully-annotated image $x_B$ and an image of a novel object $x_N$, how frequent should we apply copy-paste? There is a trade-off between oversampling and undersampling. As one extreme, one can follow Ghiasi et al. (2021) and always apply copy-paste augmentation to paste novel objects onto every base image. However, this will lead to oversampling of novel categories in the current task, which we found to hurt the performance of existing classes. On the other hand, rarely pasting novel instances would lead to undersampling of the novel class. Therefore, to guide copy-paste in the online setting, we design a synthesis strategy called vRFS based on RFS (Repeat Factor Sampling) described by Gupta et al. (2019) to perform synthesis.

To apply vRFS, we first need to compute category-wise sampling factor $r_c$ for every $c$ as in RFS. If $c \in \mathcal{C}_0$, we set $r_c = 1$ as since during the construction of $\hat{D}_0$ we already consider class balance by class-wise uniformly sampling. If $c \in \mathcal{C}_j$ with $j \geq 1$, we first compute its class frequency by $f_c = nShot/|\hat{D}_0|$, where $nShot$ denotes the number of images in $D_j$ with at least one pixel of $c$. Then, the category-wise sampling factor for $c$ is given by $r_c = \text{MAX}(1, \sqrt{t/f_c})$. Note that in (Gupta et al., 2019), $t$ is chosen as a hyperparameter to be tuned. However, we empirically found that setting $t$ to be the multiplicative inverse of total number of classes, or $t = 1/|\cup_{1,...,j} Y_j|$, is enough to yield stable results across different datasets and under different few-shot settings. This eliminates the need to search a hyperparameter for different settings and make our proposed method more robust towards different task settings.

During the synthesis process, we first randomly select a novel class $c_N$ from $\mathcal{C}_j$, and another class $c_o$ from $\cup_{1,...,j}\mathcal{C}_j \setminus \{c_N\}$. We first decide if $c_o$ should be pasted onto $x_B$. To apply vRFS resampling, we hallucinate two *virtual* samples: in the first sample where copy-paste would not be applied, the image-level sampling factor is given by $1$. In the second sample where copy-paste synthesis were to be performed, we would obtain a sample with image-level sampling factor of $r_i = \text{MAX}_{c \in i} r_c = r_{c_o}$. Thus, the probability to synthesize class $c_o$ onto $x_B$ is given by $r_{c_o}/(1 + r_{c_o})$. We then repeat the process for the novel class $c_N$. Note that vRFS synthesis is applied twice for every class, resulting in up to two pasted instances of $c_N$ in the final image.

## 4 EXPERIMENTS

### 4.1 DATASETS

We follow literature in few-shot segmentation and few-shot incremental segmentation (Shaban et al., 2017; Nguyen & Todorovic, 2019; Tian et al., 2020a; Cermelli et al., 2021) and evaluate our model on the PASCAL-5$^i$ dataset (Shaban et al., 2017) and the COCO-20$^i$ dataset (Nguyen & Todorovic, 2019). PASCAL-5$^i$ is artificially built from the PASCAL VOC 2012 Semantic Segmentation dataset (Everingham et al., 2010) with additional annotations from the SBD (Hariharan et al., 2011) dataset. The original VOC segmentation dataset provides segmentation annotations for 20 object categories. The PASCAL-5$^i$ dataset manually splits the original dataset into 4 folds for cross-validation. For each fold, 5 categories are selected as novel categories, while the remaining 15 categories are regarded as base categories. In our experiments, images containing at least one pixel of the novel categories are excluded from the base dataset. The construction of the COCO-20$^i$ dataset handles the 80 thing classes in COCO in a similar manner, where the dataset is split into 4 folds and each fold contains 20 categories. The rest of the process to construct the base dataset and the novel dataset in COCO-20$^i$ is same as the PASCAL-5$^i$ dataset.

### 4.2 EVALUATION PROTOCOLS

In the base learning stage, the model is trained using the entire base dataset. In incremental learning stages, sequences of tasks are presented to the model. We use the same evaluation protocol as proposed in Cermelli et al. (2021) for fair comparisons, where 5 incremental learning tasks are used for PASCAL-5$^i$ and each task contains 1 class from the novel split. On the COCO-20$^i$ dataset, there are 4 incremental learning tasks, and each task contains 5 classes from the novel split.

We evaluate the performance of the model on the entire validation set of the corresponding dataset after every step. For fair comparisons with Cermelli et al. (2021), we average results across different steps and exclude completely unseen classes from evaluation of current step. We use three different metrics to evaluate the performance of the model: mean Intersection-over-Union (mIoU) over base categories, mIoU over novel categories, and harmonic mean of the base mIoU and the novel mIoU. Unless otherwise noted, the numbers are computed by averaging results over splits in a cross-validating fashion.

To average out randomness due to few training samples, we also average results over multiple runs with different set of few-shot training samples. For experiments on splits on PASCAL-5$^i$, we found that averaging results over 10 runs with randomly sampled few-shot novel images yields stable results. For COCO-20$^i$, we found that averaging results over 5 runs is enough to yield stable results.

### 4.3 MAIN RESULTS

In Table 1, we evaluate various incremental segmentation methods on the PASCAL-5$^i$ dataset and the COCO-20$^i$ dataset, and combine them with GAPS where appropriate.

**Baselines.** There are two main baselines we are comparing to. The first one is SSUL (Cha et al., 2021), which is the state-of-the-art method in non-few-shot incremental segmentation. The second one is PIFS (Cermelli et al., 2021), for it is the state-of-the-art method in few-shot incremental semantic segmentation. We also report performance of our re-implementation PIFS($\mathcal{L}_2$) described in Sec. 3.1. In addition, we also evaluate simple fine-tuning and MiB (Cermelli et al., 2020).

**GAPS consistently increases performance under few-shot settings.** Methods combined with our proposed data source, GAPS, consistently outperform their un-augmented counterpart on both the base and novel categories performance. It is worth noting that GAPS substantially boosts the performance of methods that originally require fully-annotated training images (i.e., fine-tuning and PIFS), despite using only partially-annotated images now. Even for methods that do not carry out end-to-end training and update only the classifier (i.e., SSUL), GAPS still steadily increases performance on novel categories.

| METHOD | BASE | NOVEL | HM | BASE | NOVEL | HM |
|---|---|---|---|---|---|---|
| | PASCAL-5$^i$ 1-SHOT | | | PASCAL-5$^i$ 5-SHOT | | |
| MIB (Cermelli et al., 2020) | 43.9 | 2.6 | 4.9 | 60.9 | 5.8 | 10.5 |
| FINETUNE[*] | 47.2 | 3.9 | 7.2 | 58.7 | 7.7 | 13.6 |
| FINETUNE+GAPS | 64.2(+17.0) | 16.2(+12.3) | 25.9(+18.7) | 66.8(+8.1) | **38.1**(+30.4) | **48.5**(+34.9) |
| SSUL (Cha et al., 2021) | **73.9** | 16.4 | 26.8 | **74.8** | 27.8 | 40.5 |
| SSUL+GAPS | **74.0**(+0.1) | **19.9**(+3.5) | **31.3**(+4.5) | **74.9**(+0.1) | 30.0(+2.2) | 42.8(+2.3) |
| PIFS[*] (Cermelli et al., 2021) | 64.1 | 16.9 | 26.7 | 64.5 | 27.5 | 38.6 |
| PIFS($\mathcal{L}_2$)[*1] | 64.6 | 19.7 | 30.2 | 57.7 | 24.5 | 34.4 |
| PIFS($\mathcal{L}_2$)+GAPS | 66.8(+2.2) | **23.6**(+3.9) | **34.9**(+4.7) | 68.2(+10.5) | **44.2**(+19.7) | **53.7**(+19.3) |
| | COCO-20$^i$ 1-SHOT | | | COCO-20$^i$ 5-SHOT | | |
| MIB (Cermelli et al., 2020) | 40.4 | 3.1 | 5.8 | 43.8 | 11.5 | 18.2 |
| FINETUNE[*] | 38.5 | 4.8 | 8.5 | 39.5 | 11.5 | 17.8 |
| FINETUNE+GAPS | 44.5(+6.0) | **11.0**(+6.2) | 17.7(+9.5) | 46.4(+6.9) | **24.9**(+13.4) | **32.4**(+14.6) |
| SSUL (Cha et al., 2021) | **51.0** | 6.3 | 11.3 | **51.6** | 15.0 | 23.2 |
| SSUL+GAPS | **50.8**(-0.2) | **11.0**(+4.7) | **18.1**(+6.8) | **51.9**(+0.3) | 17.1(+2.1) | 25.7(+2.5) |
| PIFS[*] (Cermelli et al., 2021) | 40.4 | 10.4 | 16.5 | 41.1 | 18.3 | 25.3 |
| PIFS($\mathcal{L}_2$)[*1] | 45.7 | 10.3 | 16.8 | 46.2 | 20.2 | 28.1 |
| PIFS($\mathcal{L}_2$)+GAPS | 46.7(+1.0) | **12.8**(+2.5) | **20.1**(+3.3) | 48.8(+2.6) | **25.8**(+5.6) | **33.7**(+5.6) |

Table 1: Methods augmented with our proposed GAPS consistently outperform their un-augmented counterparts across different few-shot settings on COCO-20$^i$ and PASCAL-5$^i$. Methods noted with[*] use fully-annotated images, others use same sets of images with novel-class-only partial annotation. [1]: our re-implementation using $\mathcal{L}_2$ regularization. Highest results are colored **red** and the second highest results are colored **blue**. HM stands for harmonic mean. (Best view in color).

## 4.4 ABLATION STUDY

In Table 2, we ablate guidance designs in GAPS to illustrate how different types of guidance contribute to the final incremental learning performance than naïve copy-paste synthesis. Due to the highest harmonic mean of PIFS($\mathcal{L}_2$)+GAPS, here we use PIFS($\mathcal{L}_2$)+GAPS for the ablation study.

**Our diversity-guided exemplar selection method consistently increases performance on base categories**, which suggests that it is capable of choosing diverse samples to construct a representative memory-replay buffer and mitigate catastrophic forgetting.

**Context-guided sampling steadily improves performance on novel classes**, which is consistent with findings in previous work (Dvornik et al., 2018) that background context is an important factor to consider in copy-paste synthesis.

**Frequency-guided probabilistic synthesis boosts results on novel classes**. On the other hand, its influence on base categories is negligible. We take a closer look at step-wise performance (whose visualization is available in Appendix A.11) and found that the reason is due to unguided copy-paste's oversampling of novel classes that are being adapted, and forgetting of classes learned in the previous incremental learning stage and not in the memory-replay buffer.

## 4.5 MORE CHALLENGING SINGLE-INSTANCE EXPERIMENT

Though the semantic background shift proposed by Cermelli et al. (2020) relaxes the requirement to provide full segmentation annotations, it still requires *all novel instances* in images to be annotated, which can be time-consuming to obtain in cluttered scenes and hinder potential applications. Here we consider a more challenging task setting, which we term *single-instance incremental learning*. Namely, for training images provided in incremental learning stages, if there are multiple instances of a novel class in the image, we assume that only *one instance* will be annotated.

To simulate this setting, we use the instance-level segmentation annotation provided by the COCO dataset to enforce only annotation of one novel instance in every image is available to the model. Since state-of-the-art incremental segmentation approaches use pseudo-labeling (Cha et al., 2021), we design a method PIFS($\mathcal{L}_2$)$^\dagger$, which simulates combining PIFS($\mathcal{L}_2$) with pseudo-labeling to cope with partially-annotated sample. Here we allow PIFS($\mathcal{L}_2$)$^\dagger$ to have access to additional information

| MEM | COPY-PASTE | F-GUIDE | D-GUIDE | C-GUIDE | BASE | NOVEL | HM |
|---|---|---|---|---|---|---|---|
| — | —* | — | — | — | $46.2_{(\pm 0.3)}$ | $20.2_{(\pm 0.7)}$ | $28.1_{(\pm 0.3)}$ |
| ✓ | —* | — | — | — | $49.3_{(\pm 0.2)}$ | $19.4_{(\pm 0.7)}$ | $27.9_{(\pm 0.3)}$ |
| ✓ | ✓ | — | — | — | $47.0_{(\pm 0.2)}$ | $19.8_{(\pm 0.6)}$ | $27.8_{(\pm 0.3)}$ |
| ✓ | ✓ | ✓ | — | — | $47.2_{(\pm 0.2)}$ | $25.2_{(\pm 0.6)}$ | $32.9_{(\pm 0.3)}$ |
| ✓ | ✓ | ✓ | ✓ | — | $48.2_{(\pm 0.2)}$ | $25.0_{(\pm 0.7)}$ | $32.9_{(\pm 0.3)}$ |
| ✓ | ✓ | ✓ | ✓ | ✓ | $\mathbf{48.8}_{(\pm 0.2)}$ | $\mathbf{25.8}_{(\pm 0.6)}$ | $\mathbf{33.7}_{(\pm 0.3)}$ |

Table 2: Ablation study of components in GAPS on PIFS($\mathcal{L}_2$) on the COCO-20$^i$ dataset under 5-shot setting. Note that when only combined with the memory-replay buffer, the base IoU is higher because model has access to additional full annotations. When diversity guidance (D-guide) is disabled, $\hat{D}_0$ consists of random examples from the base dataset, resulting in worse base performance. When context guidance (C-guide) is disabled, a base image is uniformly sampled. When frequency guidance (F-guide) is disabled, a novel instance is sampled uniformly and is always pasted onto the base image. 95% confidence intervals over 20 trials are reported assuming that trial results are normally distributed. *: use fully-annotated masks when copy-paste is turned off.

| METHOD | BASE | NOVEL | HM | BASE | NOVEL | HM |
|---|---|---|---|---|---|---|
| | | ALL INSTANCES | | | SINGLE-INSTANCE ONLY | |
| PIFS($\mathcal{L}_2$)$^†$ | 46.2 | 20.2 | 28.1 | 46.1 (-0.2%) | 17.6 (-12.9%) | 25.4 (-9.6%) |
| PIFS($\mathcal{L}_2$)+GAPS | 48.8 | 25.8 | 33.7 | 49.1 **(+0.6%)** | 25.1 **(-2.7%)** | 33.2 **(-1.5%)** |

Table 3: Performance of pseudo-labeling methods and GAPS under the more challenging single-instance learning setting on COCO-20$^i$ 5-shot. Only 1 novel instance out of potentially many instances in individual training images is annotated. The pseudo-labeling baseline, PIFS($\mathcal{L}_2$)$^†$, yields substantially worse performance; whereas PIFS($\mathcal{L}_2$)+GAPS has only minor performance decreases.

– the annotation of other non-novel background pixels – to simulate an oracle pseudo-labeling model which perfectly segments learned classes but recognizes unseen novel classes as background.

The results are given in Table 3. We can observe that the pseudo-labeling baseline, PIFS($\mathcal{L}_2$)$^†$, yields substantially worse performance when the model receives single-instance annotations. We reason this is due to noisy labels generated by the pseudo-labeling process, where novel instances are incorrectly labeled as background. On the contrary, PIFS($\mathcal{L}_2$)+GAPS shows only a minor performance decrease with single instances. This highlights the potential of copy-paste synthesis as an alternative to the existing pseudo-labeling paradigm to cope with the more realistic single-instance setting.

**More results and visualization.** Due to space limit, we kindly refer readers to the appendix for more quantitative results from Appendix A.2 and visualized qualitative results in Appendix A.11.

## 5 CONCLUSION AND DISCUSSION

In this paper, we demonstrate how judicious use of copy-paste dramatically boosts the performance of incremental segmentation methods under the few-shot setting and enables learning with partially-annotated images. Our proposed GAPS technique selects representative exemplars in the memory-replay buffer and addresses the problems of class imbalance and contextual mismatch in synthesis.

In future work, we are interested in further application of copy-paste as a synthesis technique to cope with the background shifting problem for incremental segmentation. We believe that copy-paste can serve as a promising alternative to pseudo-labeling and modified loss to enable learning on partially-annotated images. We also believe that further optimizing exemplar selection and sampling strategies can lead to better guidance and lead to even better performance. Finally, the ability to learn with as few as one annotated instance in an image raises several intriguing possibilities. For example, integrating our work with learning-based interactive segmentation will enable human operators to continually and adaptively teach novel classes and correct failed predictions. This workflow has many interesting applications such as robot teleoperation where sparse annotations are preferable. Learning with weaker annotations, like bounding boxes or single clicks, and even self-supervision, is also an interesting direction to explore.

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

| Split | Categories |
|-------|------------|
| COCO-20-0 | person, airplane, boat, parking meter, dog
elephant, backpack, suitcase, sports ball, skateboard
wine glass, spoon, sandwich, hot dog, chair
dining table, mouse, microwave, refrigerator, scissors |
| COCO-20-1 | bicycle, bus, traffic light, bench, horse
bear, umbrella, frisbee, kite, surfboard
cup, bowl, orange, pizza, couch
toilet, remote, oven, book, teddy bear |
| COCO-20-2 | car, train, fire hydrant, bird, sheep
zebra, handbag, skis, baseball bat, tennis racket
fork, banana, broccoli, donut, potted plant
tv, keyboard, toaster, clock, hair drier |
| COCO-20-3 | motorcycle, truck, stop sign, cat, cow
giraffe, tie, snowboard, baseball glove, bottle
knife, apple, carrot, cake, bed
laptop, cell phone, sink, vase, toothbrush |

Table 4: COCO-20$^i$ class splits under multi-step settings. Every novel split is further split into 4 steps sequentially sorted by their original class indices as in (Cermelli et al., 2021). A row indicates classes in a step. For example, in COCO-20-1, classes presented to the model in the first incremental learning step are bicycle, bus, traffic light, bench, and horse.

| Split | Categories |
|-------|------------|
| PASCAL-5-0 | aeroplane \| bicycle \| bird \| boat \| bottle |
| PASCAL-5-1 | bus \| car \| cat \| chair \| cow |
| PASCAL-5-2 | table \| dog \| horse \| motorbike \| person |
| PASCAL-5-3 | potted plant \| sheep \| sofa \| train \| tv-monitor |

Table 5: PASCAL-5$^i$ class splits under multi-step settings. Every novel split is further split into 5 steps sequentially sorted by their original class indices and each step is split by the '|'.

# A APPENDIX

## A.1 DATASET SPLITS

We use the PASCAL-5$^i$ splits (Shaban et al., 2017) and the COCO-20$^i$ splits (Nguyen & Todorovic, 2019) as described in previous works. We follow the multi-step setup described in (Cermelli et al., 2021) to ensure fair comparisons. Details are given in Table 4 and Table 5.

## A.2 MEMORY-REPLAY BUFFER CONSTRUCTION

In Table 6, we study how different memory buffer construction strategies influence the performance of the model. We experiment with multiple baseline implementations to construct the memory-replay buffer: *Full-set-random* denotes samples that are uniformly sampled from the base training set. *Classwise-random* denotes randomly selecting even number of samples from each base class. To consider multiple classes in images, we design two baseline methods. RFS (Repeat Factor Sampling) is done by using the weighted sampling technique described in LVIS (Gupta et al., 2019), which considers multiple classes in samples by computing category-level re-sampling factor and creates image-level re-sampling factor based on the maximum category-level factor in the image. RFS assigns higher weights to rare classes and has been shown to work well in long-tailed task settings. The other baseline, 'GreedyClass', is achieved by greedily picking samples in every base class with the most number of classes. To consider the region sizes in the image, we design a baseline method, 'BalancedRegion', which uses a similar algorithm as our proposed diversity sampling method, but we replace the difficulty estimation with the number of pixels the selected class occupies in images.

| Base Set | Base | Novel | HM |
|---|---|---|---|
| Full-set-random | $47.9 \pm 0.2$ | $24.6 \pm 0.7$ | 32.5 |
| Classwise-random | $48.3 \pm 0.2$ | $24.7 \pm 0.8$ | 32.7 |
| RFS | $47.6 \pm 0.3$ | $25.2 \pm 0.7$ | 33.0 |
| GreedyClass | $48.7 \pm 0.2$ | $25.0 \pm 0.7$ | 33.0 |
| BalancedRegion | $48.6 \pm 0.2$ | $24.9 \pm 0.8$ | 32.9 |
| Easy | $48.0 \pm 0.2$ | $24.2 \pm 0.8$ | 32.2 |
| Hard | $47.7 \pm 0.2$ | $25.2 \pm 0.8$ | 33.0 |
| Easy-Hard-Mix | $48.4 \pm 0.3$ | $24.8 \pm 0.7$ | 32.8 |
| Diverse Sampling (Ours) | $\mathbf{48.9} \pm 0.2$ | $\mathbf{25.3} \pm 0.9$ | **33.3** |

Table 6: Ablation study of different memory-replay buffer construction strategies on COCO-20-1. We found that the performance is maximized when using our diversity-aware memory-replay buffer reconstruction strategy. Mean and 95% confidence intervals over 10 runs are reported.

| Method | Base | Novel | HM |
|---|---|---|---|
| Always Paste (Ghiasi et al., 2021) | $47.1 \pm 0.3$ | $18.7 \pm 1.7$ | 26.8 |
| 50-50-Paste | $\mathbf{49.7} \pm 0.4$ | $24.2 \pm 1.5$ | 32.6 |
| CAS (Gupta et al., 2019) | $49.4 \pm 0.3$ | $12.0 \pm 1.3$ | 19.3 |
| vRFS (Ours) | $48.9 \pm 0.4$ | $\mathbf{25.3} \pm 1.6$ | **33.3** |

Table 7: Ablation study of different strategies to compute synthesis probability on COCO-20-1. Our proposed vRFS method computes synthesis probability to balance class frequencies in training data, and achieves best performance in terms of harmonic mean. Mean and standard deviation over 10 runs are reported.

Intuitively, easy samples help model maintain learned prototypes during incremental learning, and hard samples help model distinguish decision boundaries. Hence, based on our sample difficulty ranking method based on prototypical distances, we also construct a few variants: *Easy* denotes samples whose predicted embeddings are closest to learned prototypes; *Hard* represents difficult samples whose predicted embedding are most distant to learned prototypes in the learned metric space; *Easy-Hard-Mix* denotes equal mixtures of samples drawn from easiest samples and hardest samples. Lastly, *Diverse-Sampling* is our proposed method, where samples of all difficulties are equally drawn.

We experimented with all of these variants and found that the diversity-aware construction yields the best result in all metrics. Compared to randomly selected samples or easy samples, using hard samples result in better performance on novel categories. On the other hand, we found that using a mixture of easiest samples and hardest sample is also a good strategy, which achieves superior performance in base categories compared to other baselines, and comparable performance to our diversity-aware sampling. It is also worth noting that both the 'GreedyClass' method and the 'BalancedRegion' method achieve almost comparable results to our methods, but still slightly underperform our diversity guidance.

### A.3 CLASS-FREQUENCY-AWARE SYNTHESIS

We investigate how different probabilistic strategies to apply copy-and-paste augmentation affects GAPS. The results are illustrated in Table 7. Here, *vRFS* (virtual RFS) is our proposed method. *CAS* (Class-aware-sampling) is a sampling strategy from (Gupta et al., 2019) which first uniformly samples a class and then selects a sample containing instances from the selected class. In our implementation, the probability to synthesize novel objects is simply given by $1/M$, where $M$ is the sum of total number of base classes and seen novel classes. For every selected base image, the *Always Paste* method randomly picks a novel category and always synthesizes an instance of the selected novel category onto the given image. Finally, the *50-50-Paste* method is an extension of the naïve copy-paste method to leverage past novel samples like vRFS. Essentially, it replays samples from learned novel classes with 50% probability, and uses novel class from the current incremental learning task also with 50% probability. Note that these two options are not mutually exclusive.

| Method | Base | Novel |
|---|---|---|
| Standard Jittering | $48.5 \pm 0.4$ | $24.1 \pm 1.5$ |
| Large Scale Jittering (Ghiasi et al., 2021) | $\mathbf{48.9 \pm 0.4}$ | $\mathbf{25.3 \pm 1.5}$ |

Table 8: Ablation study of magnitudes of random resizing in copy-paste on COCO-20-1. Large Scale Jittering is proposed in (Ghiasi et al., 2021) with more aggressive resizing scale. Mean and standard deviation over 10 runs are reported.

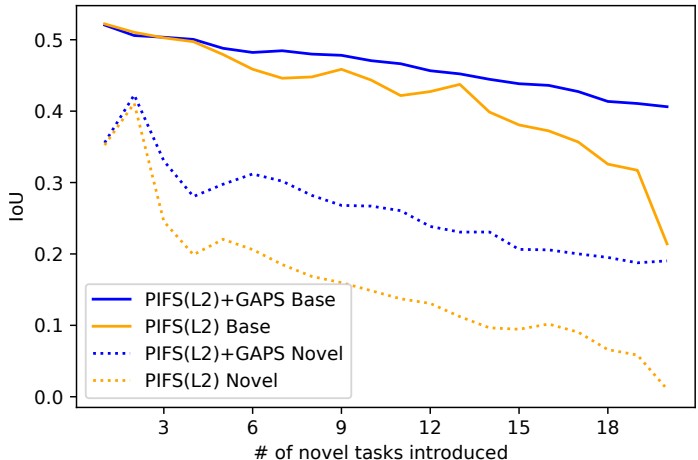

Figure 3: Long-term continual learning performance of PIFS($\mathcal{L}_2$) and PIFS($\mathcal{L}_2$)+GAPS on the COCO-20-1 split under 5-shot setting. Best view in color.

In the result, we can observe that our method (vRFS) achieves the best result in terms of the novel IoU and harmonic mean of base and novel IoU. On the other hand, the 50-50-Paste baseline and the CAS baseline achieves better results possibly because they undersample novel classes.

## A.4   EFFECT OF GEOMETRIC VARIATIONS

In Table 8, we investigate how geometric variations affect the performance in our method. Our conclusion is consistent with what was found in (Ghiasi et al., 2021): using more aggresive scale jittering helps increase the diversity of samples and increases performance. Beyond confirming conclusion from previous work, such improvement alongside stronger augmentation demonstrates the effectiveness of augmentation technique even when there is few data presented, highlighting a possibility to further improve the performance by using even stronger augmentation technique.

## A.5   ROBUSTNESS AGAINST CATASTROPHIC FORGETTING

We perform an additional experiment on COCO-20-1 where a long few-shot incremental learning sequence (60 base; 20 novel classes with 20 incremental steps; 5-shot) is used. The step-wise result can be found in Fig. 3. PIFS($\mathcal{L}_2$), augmented with GAPS, demonstrates more robustness to forgetting with many incremental learning steps than standard PIFS.

## A.6   EVALUATION ON DISJOINT ADE20K 100-5 5-SHOT SETTING

To validate our approach on the ADE20k dataset (Zhou et al., 2017b; 2019), we ran a suite of experiments where we follow the ADE100-5 (11 steps) setup discussed in (Cha et al., 2021) and modify it as 5-shot learning. We tested PIFS($\mathcal{L}_2$) and PIFS($\mathcal{L}_2$)+GAPS on this setting. The results are given in Table 9. We observe a steady increase in both the base IoU and novel IoU in this challenging task setting on methods augmented with GAPS. This further validates the effectiveness of our approach.

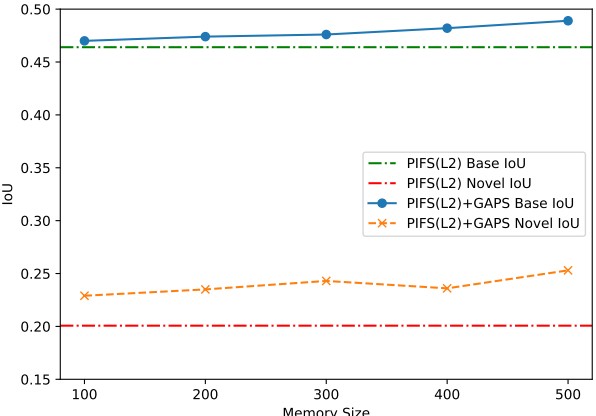

Figure 4: Performance of PIFS($\mathcal{L}_2$)+GAPS on the COCO-20-1 split under 5-shot setting with varying memory-replay buffer size. Best view in color.

| Method | Base | Novel | HM |
|---|---|---|---|
| PIFS($\mathcal{L}_2$) | $33.8 \pm 0.9$ | $3.5 \pm 1.5$ | 6.3 |
| PIFS($\mathcal{L}_2$)+GAPS | $36.2 \pm 0.6$ | $5.9 \pm 1.4$ | 10.1 |

Table 9: Performance of PIFS($\mathcal{L}_2$) and PIFS($\mathcal{L}_2$)+GAPS on the ADE100-5 disjoint 5-shot setting. Mean and standard deviation over 5 runs are reported.

## A.7 COMPARISONS WITH EXEMPLAR-ONLY METHODS

We explore an alternative memory-replay buffer implementation to add examples learned during the incremental learning stage to the memory-replay buffer, in an attempt to help performance on novel classes. Since examples observed during the incremental learning stage are only partially annotated, they can not be directly added to the memory-replay buffer. We experiment with an alternative method. Here, 'PIFS($\mathcal{L}_2$)+MEM+PL' means that we follow SSUL and perform pseudo-labeling on partially annotated samples before adding them to memory. The results are given in Table 10. We can observe that storing samples observed during the incremental learning stage leads to only moderate increase in novel IoU. Our PIFS($\mathcal{L}_2$)+GAPS significantly outperforms both baselines.

## A.8 EFFECT OF MODELS PRETRAINED ON DIFFERENT DATASETS FOR CONTEXTUAL GUIDANCE

We experiment with several other pretraining models as the contextual guiding model. In particular, we use ImageNet-pretrained model and pretrained image encoder from CLIP (Radford et al., 2021) to compute scene embedding for incoming images. The other settings remain unchanged. The results are given in Table 11. We observe from the results that the ImageNet-pretrained model performs worse than our Places365-pretrained model. On the other hand, the CLIP-pretrained model yields comparable performance to our method.

In summary, we conclude that Places365 is an appropriate dataset for training the image-level contextual sampler from both quantitative. Qualitative discussions can be found in Figure 6.

## A.9 PERFORMANCE WITH VARYING MEMORY SIZES

We conducted an additional experiment on the COCO-20-1 dataset under the 5-shot setting, where we vary the number of fully-annotated images in the memory-replay buffer to investigate the per-

| Method | Base | Novel | HM |
|---|---|---|---|
| PIFS($\mathcal{L}_2$) | $46.4 \pm 0.2$ | $20.7 \pm 1.0$ | 28.0 |
| PIFS($\mathcal{L}_2$)+MEM | $49.4 \pm 0.2$ | $18.6 \pm 0.9$ | 27.0 |
| PIFS($\mathcal{L}_2$)+MEM+PL | $48.4 \pm 0.2$ | $21.5 \pm 1.0$ | 29.8 |
| PIFS($\mathcal{L}_2$)+GAPS | $48.9 \pm 0.2$ | $25.3 \pm 0.9$ | 33.3 |

Table 10: Comparisons of GAPS and various exemplar-only methods with PIFS($\mathcal{L}_2$). Mean and 95% confidence intervals over 10 runs are reported.

| Pretrained Model | Base | Novel | HM |
|---|---|---|---|
| ImageNet | $48.2 \pm 0.2$ | $24.3 \pm 0.9$ | 32.3 |
| CLIP | $48.5 \pm 0.2$ | $25.4 \pm 1.0$ | 33.3 |
| Places365 | $48.9 \pm 0.2$ | $25.3 \pm 0.9$ | 33.3 |

Table 11: Performance different pretrained models for contextual guidance under the COCO-20-1 5-shot setting. Mean and 95% confidence intervals over 10 runs are reported.

formance of our method. As can be seen from Fig. 4, even with as few as 100 fully-annotated base images, GAPS still consistently boosts the performance of the underlying PIFS($\mathcal{L}_2$) method.

## A.10 IMPLEMENTATION DETAILS

To allow fair comparisons with existing methods (Cermelli et al., 2021; Cha et al., 2021; Cermelli et al., 2020), we use the same Deeplab-V3 (Chen et al., 2017) architecture with ResNet-101 (He et al., 2016) backbone. Following existing works in incremental few-shot segmentation literature (Cermelli et al., 2021; Tian et al., 2020a), before the beginning of base learning stage, the ResNet-101 backbone is initialized using weights pre-trained on ImageNet.

For fine-tuning and PIFS($\mathcal{L}_2$), during the base training stage, we use an initial learning rate of 0.007 and polynomial learning rate schedule with batch size of 32 and train for 20 epochs on both the PASCAL-$5^i$ dataset and the COCO-$20^i$ dataset. SGD optimizer is used with 0.9 momentum coefficient and 0.0001 weight decay. During the incremental learning stage, we apply initial learning rates of 0.001 and 0.01 for the feature extractor and the classifier, respectively. Polynomial learning rate schedule is used. For the PASCAL-$5^i$ dataset, we use batch size of 16 for 200 iterations and $\lambda = 0.1$. For the COCO-$20^i$ dataset, we use batch size of 16 for 400 iterations and $\lambda = 0.5$. 100 iterations are used for 1-shot cases on COCO. For data augmentation, we follow implementations in PIFS (Cermelli et al., 2021) and SSUL (Cha et al., 2021) and use standard random horizontal flipping, random resizing, and random cropping to $512 \times 512$.

The reproduction of SSUL and the re-implementation SSUL+GAPS uses the same set of hyperparameters and augmentations from the open-source release of SSUL except for the batch size which is changed to 16.

The scene classification network is the original VGG16 (Simonyan & Zisserman, 2015) network without batch normalization layers. We replace the last linear classification layer with a prototypical learning layer to make the network generate unit vector embedding for comparison using cosine similarity. The network is trained on the training split of the Places365 dataset (Zhou et al., 2017a) for 6 epochs. We use initial learning rate of 0.01 and polynomial learning rate schedule with batch size of 64. All images are bilinearly interpolated to size $224 \times 224$. The final scene classification network achieves 0.51 accuracy on the validation split of the Places365 dataset.

## A.11 QUALITATIVE RESULTS

Fig. 5 shows qualitative results of PIFS($\mathcal{L}_2$), PIFS($\mathcal{L}_2$)+NPS (naive copy-paste), and PIFS($\mathcal{L}_2$)+GAPS on the PASCAL-5-1 split dataset under the 1-shot setting using images on the left column. We can see that PIFS and PIFS+NPS gives reasonably good predictions on classes that are adapted recently (e.g., the cow class in the last row). However, they tend to exhibit catastrophic forgetting for classes learned earlier (e.g., the bus example in PIFS). On the other hand, the

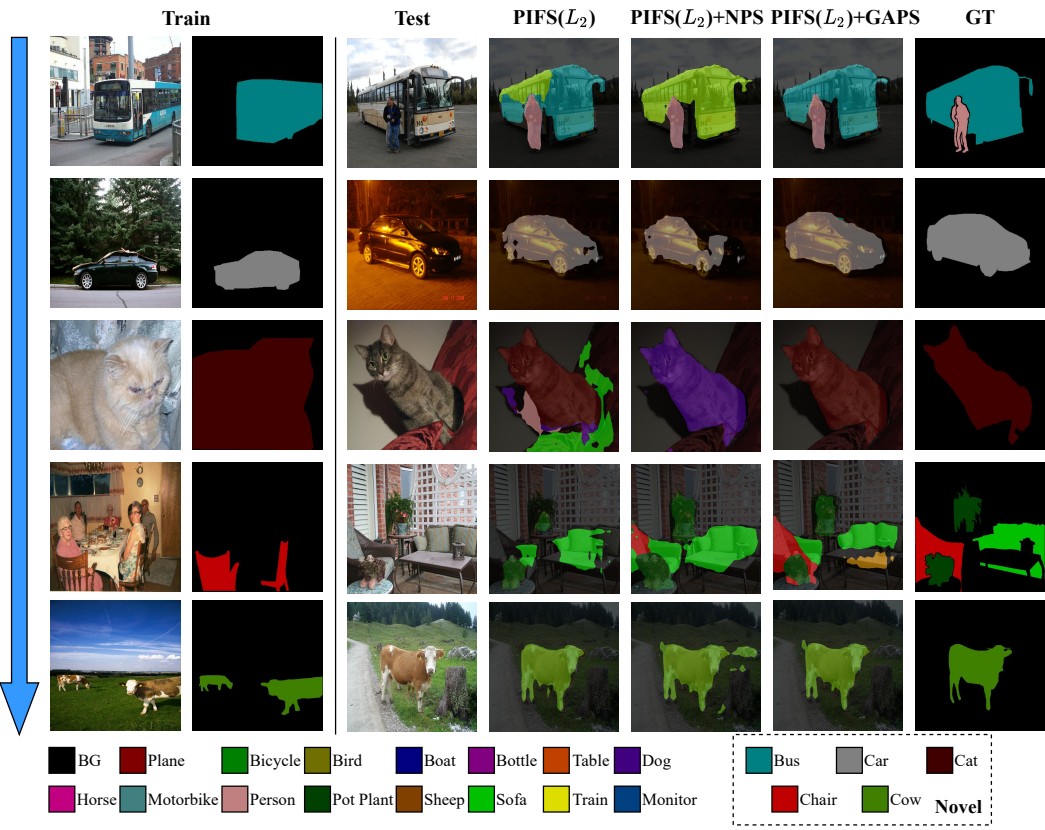

Figure 5: Qualitative results on the PASCAL-5-1 split under 1-shot setting. Sequential adaptation to five classes are performed in the order indicated by the blue arrow on the left. Visualization is done after all adaptation steps are completed. Observe, particularly in rows 1, 3, and 4, that our method mitigates catastrophic forgetting and improves performance on both base and novel classes compared to PIFS and PIFS+NPS. (Best viewed in color)

model trained using GAPS not only yields good result on recently learned classes, but also maintains promising performance on classes adapted earlier.

In Fig. 6, we give some qualitative examples of the scene embedding network. We can see that given an image of traffic lights on a railway, our scene embedding model is able to select contextually-similar scenes such as railways and street scenes for subsequent synthesis.

Fig. 7 and Fig. 8 demonstrate how PIFS($\mathcal{L}_2$)+GAPS compare to the baseline PIFS($\mathcal{L}_2$) across multiple step. The set up is the same as Pascal-5$^{\text{i}}$ 1-shot incremental learning. These visualizations demonstrate how PIFS($\mathcal{L}_2$) forgets learned classes earlier as more incremental learning steps occur. On the other hand, PIFS($\mathcal{L}_2$)+GAPS is able to maintain good performance on learned classes as it adapts to new classes.

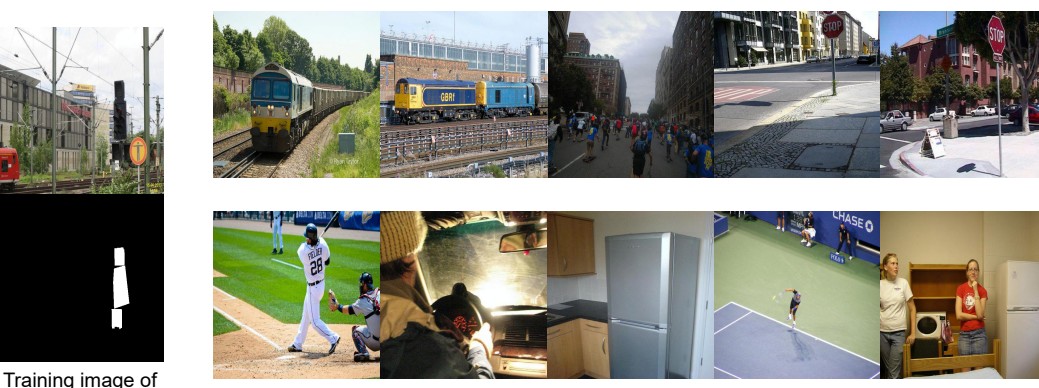

Figure 6: Qualitative visualization of samples selected in the context-aware sampling process. Given an image of traffic lights in railway, our scene embedding network is able to select contextually-similar scenes (railway, street).

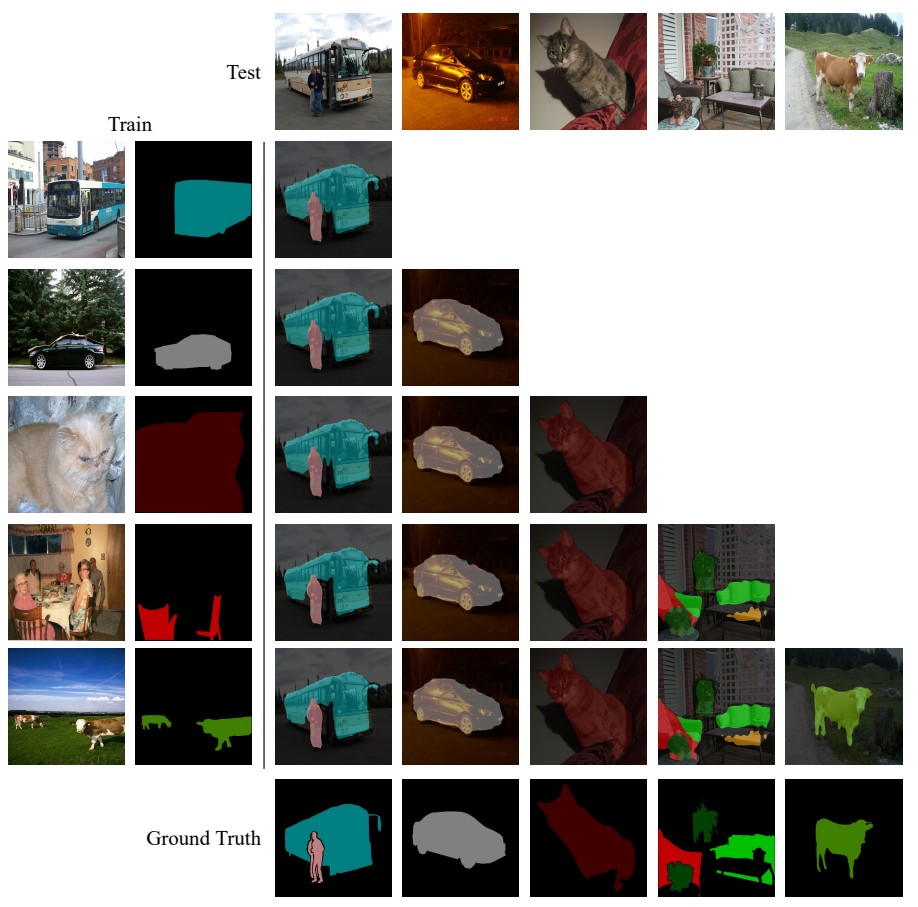

Figure 7: Multi-step qualitative results of our method (GAPS) on the PASCAL-5-1 split under 1-shot setting. (Best view in color.)

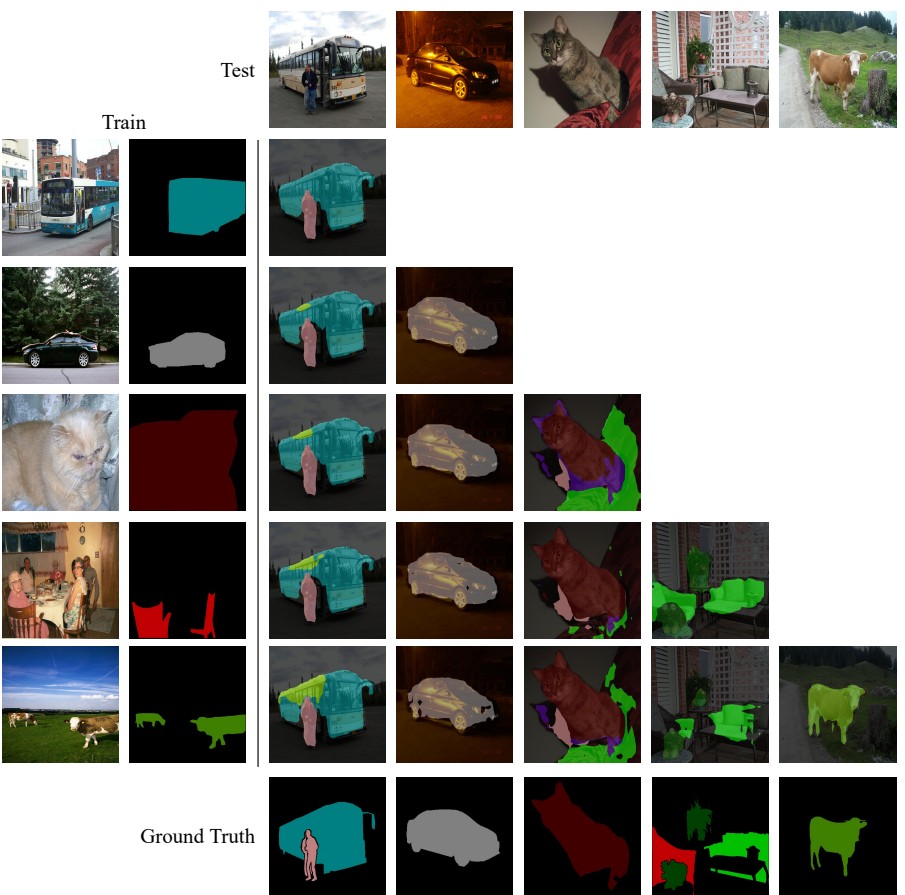

Figure 8: Multi-step qualitative results of PIFS($\mathcal{L}_2$) on the PASCAL-5-1 split under 1-shot setting. (Best view in color.)

