# OpenReview forum: "GAPS: Few-Shot Incremental Semantic Segmentation via Guided Copy-Paste Synthesis"
_ICLR.cc/2023/Conference — Submitted to ICLR 2023_

### Official Review · Reviewer_2ZbE · 2022-10-21

**Confidence:** 5
**Correctness:** 3
**Technical Novelty And Significance:** 3
**Empirical Novelty And Significance:** 3
**Recommendation:** 5

**Clarity, Quality, Novelty And Reproducibility:**

The paper is well-written and properly structured.
The paper provides multiple details to reproduce the code.



**Strength And Weaknesses:**

Strengths:
1. The paper investigates a novel and understudied scenario that is more realistic than earlier benchmarks.
2. The study gives a comprehensive summary of similar works as well as comparisons and contrasts between them.
3. The study expands the present benchmark with new experiments involving only one labeled instance of the new classes, demonstrating that the GAPS can address the issue of missing annotations.

Weaknesses:
1. The ablation study may be extended to other settings and may provide more insights about the paper choices. In particular, it may start from a baseline that uses exemplars without pasting the new classes on them, to show that pasting is important to improve performance and that the gain is not only given by the regularization effect of using exemplars (as proposed by GDumb [a]). Moreover, the paper should clarify if, while disabling the diversity guidance, the exemplars are sampled uniformly for the old classes or if they are randomly sampled on the full dataset.
2. From the ablation study, context sampling appears to marginally improve the results. Moreover, it is not clear how the performance will change by using a pre-training dataset different from Places-365, such as ImageNet or ADE20K. Furthermore, can it be trained directly on the base dataset of the considered setting or can the method use the segmentation network itself (for example, using an auxiliary classifier on the encoder features)?
3. The paper baselines appear to be very naive. In particular, the paper reports the results for Fine-Tuning, MiB, and PIFS without considering exemplars, introducing an unfair comparison with respect to GAPS. The baselines should be reported both without exemplars (as in the original paper) and with them (indeed, without the proposed copy-and-paste strategy).

[a] Prabhu, Ameya, Philip HS Torr, and Puneet K. Dokania. "Gdumb: A simple approach that questions our progress in continual learning." European conference on computer vision. Springer, Cham, 2020.

**Summary Of The Paper:**

The paper proposes a data augmentation technique to improve the performance of the existing state-of-the-art methods in the incremental few-shot semantic segmentation setting.
The proposed technique, called GAPS, augments the few-shot samples by pasting the new classes' pixels onto a subset of exemplar images coming from the previous training steps. There are three critical aspects of this technique. First, exemplar selection: the subset of exemplar images is selected by uniformly sampling the images considering the similarity with the class prototype. Second, where to paste the new classes: the images are selected based on the similarity to the context of the new classes ranked with a VGG pre-trained on Places-365. Third, how frequently apply copy and paste: the paper proposes a dynamic strategy called virtual Repeated Factor Sampling.
The results show that GAPS is able to improve state-of-the-art methods by a substantial margin on both Pascal-5 and COCO-20 settings, either using one or five shots.


**Summary Of The Review:**

Overall, the paper proposes a new interesting data-augmentation technique for exemplar-based Incremental Few-Shot semantic segmentation. However, the paper misses some important comparisons, both in the ablation studies and in the main tables.

---

> ### Author Response · Authors · 2022-11-19
> **Response for Reviewer 2ZbE (part 1/2)**
>
> We thank the reviewer for the valuable comments. Below we address all the concerns.
>
> ### 1. Ablation study on baseline with exemplars without copy-pasting / Stronger baseline
>
> We thank the reviewer for the insightful suggestions. This response serves as an answer to the first and third points listed in ‘weaknesses,’ as both comments are concerned about results with exemplars but without GAPS. We perform **two additional suites of experiments** to answer the questions: 1) we add an **additional entry (second row) in the ablation table where exemplars are used without copy-pasting**, and 2) we set up a few more baselines by implementing different memory construction strategy to combine with PIFS(L2) to illustrate how it performs with exemplars but without copy-paste. In both suites of experiments, our proposed GAPS method significantly outperforms baseline implementations.
>
> - Specifically, In the first experiment, we added an additional entry ‘PIFS(L2)+MEM’ (second row) where the same set of exemplars as GAPS are used during incremental learning, but without copy-pasting. We can observe that learning with additional exemplars from the base dataset increases performance on base classes, but slightly decreases novel performance. The combined harmonic mean performance is significantly worse than GAPS.
>
> | Method  | Base | Novel | HM |
> |  ----------- | -----------    | ----------- | ----------- |
> | PIFS(L2)  | 46.2 +- 0.3 | 20.2 +- 0.7 | 28.1 |
> | PIFS(L2)+MEM | 49.3 +- 0.2 | 19.4 +- 0.7 | 27.9 |
> | PIFS(L2)+GAPS | 48.8 +- 0.2 | 25.8 +- 0.6 | 33.7 |
>
>
> - In the second experiment, we explore an alternative memory-replay buffer implementation to add examples learned during the incremental learning stage to the memory-replay buffer, in an attempt to help performance on novel classes. Since examples observed during the incremental learning stage are only partially annotated, they can not be directly added to the memory-replay buffer. We experiment with an alternative method. Here, ’PIFS(L2)+MEM+PL‘ means that we follow SSUL and perform pseudo-labeling on partially annotated samples before adding them to memory. We can observe that storing samples observed during the incremental learning stage leads to only moderate increase in novel IoU. Our PIFS(L2)+GAPS significantly outperforms both baselines.
>
> | Method  | Base | Novel | HMl |
> |  ----------- | -----------    | ----------- | ----------- |
> | PIFS(L2)+MEM | 49.4 +- 0.2 | 18.6 +- 0.9 | 27.0 |
> | PIFS(L2)+MEM+PL | 48.4 +-0.2  | 21.5 +- 1.0 | 29.8 |
> | PIFS(L2)+GAPS | 48.9 +- 0.2 | 25.3 +- 0.9 | 33.3 |
>
> ### 2. Clarifications for diversity-guided ablation
>
> We would like to clarify that, in the ablation study, when the diversity guidance is disabled, the memory-replay buffer is constructed by random sampling of the full dataset. To illustrate that random sampling uniformly from old classes does not outperform our diversity guidance, we added an additional experiment ‘Classwise-random’ in Table 6 in the Appendix to demonstrate the performance where exemplars are selected uniformly for old classes. We list the results here for easy reference. More details and discussions can be found in Appendix Sec. A.2 and Table 6 in the updated paper PDF.
>
> | Method  | Base | Novel | HM |
> |  ----------- | -----------    | ----------- | ----------- |
> | Full-set-random | 47.9 +- 0.2 | 24.6 +- 0.7 | 32.5 |
> | Classwise-random | 48.3 +- 0.2 | 24.7 +- 0.8 | 32.7 |
> | DiverseSampling | 48.9 +- 0.2 | 25.3 +- 0.9 | 33.3 |

---

> > ### Author Response · Authors · 2022-11-19
> > **Response for Reviewer 2ZbE (part 2/2)**
> >
> > ### 3. Contextual Guidance
> >
> > - Thanks for the suggestion. First, we will discuss the benefit of proposed contextual guidance both quantitatively and qualitatively. Quantitatively, we have updated the ablation study table, and demonstrate that contextual guidance raises performance in a statistically significant manner above the 95% confidence interval. We include the results here for easy reference. Qualitatively, we refer the reviewer to figure 5 in the appendix, which shows how our contextual guidance generates more contextually meaningful images.
> >
> > - Regarding varying datasets - we experiment with several other pretraining models as the contextual guiding model. In particular, we use ImageNet-pretrained model and pretrained image encoder from CLIP [1] to compute scene embedding for incoming images. The other settings remain unchanged. We observe from the results that the ImageNet-pretrained model performs worse than our Places365-pretrained model. On the other hand, the CLIP-pretrained model yields comparable performance to our method.
> >
> > | Pretrained Model  | Base | Novel | HM |
> > |  ----------- | -----------    | ----------- | ----------- |
> > | ImageNet  | 48.2 | 24.3 | 32.3 |
> > | CLIP | 48.5 | 25.4 | 33.3 |
> > | Places365 | 48.9 | 25.3 | 33.3 |
> >
> > In summary, we conclude that Places365 is an appropriate dataset for training the image-level contextual sampler from both qualitative and quantitative evidence.
> >
> > - Regarding training an auxiliary classifier on top of the encoder feature - while it is possible to do so after the base learning stage, during the incremental learning stage, the encoder may be fine-tuned to adapt to novel classes, which can lead to catastrophic forgetting of this auxiliary branch. Conceptually, maintaining the performance of this auxiliary branch would require additional multi-task consideration, which we would like to leave to future work.
> >
> > [1] Radford, Alec, et al. "Learning transferable visual models from natural language supervision." ICML. 2021.

---

> > > ### Comment · Area_Chair_S9Mv · 2022-11-23
> > > **Following up on rebuttal**
> > >
> > > Dear reviewer,
> > >
> > > the authors tried to address your comments, most notably on the account of clarity and naivete of experimental settings. Has the rebuttal in general influenced your view?
> > >
> > > Cheers, Your AC

---

> > > > ### Comment · Reviewer_2ZbE · 2022-11-25
> > > > **Following up on rebuttal**
> > > >
> > > > Dear AC, dear authors,
> > > >
> > > > The rebuttal extensively addressed my concerns, reporting experimental results for all the review points.
> > > > The paper clearly improved reporting stronger baselines and better comparisons.
> > > >
> > > > However, Contextual Guidance is still not convincing. Although it slightly improves the performance, it requires training an additional network on an external dataset, which choice may influence the final performance.

---

> > > > > ### Author Response · Authors · 2022-11-28
> > > > > **Response to follow-up questions on contextual guidance**
> > > > >
> > > > > We thank the reviewer for the response. And it is delightful to see that the additional reported results have addressed most of the reviewer's concerns.
> > > > >
> > > > > ## Q1: Contextual guidance
> > > > >
> > > > > To further address the remaining concern that contextual guidance requires the training of an additional network on an external dataset, we experiment with contextual guidance that directly uses the segmentation network itself on the base dataset of the considered setting, following the original suggestion from the reviewer. Interestingly, **this experiment shows that contextual guidance does not need to require an additional network or an external dataset**.
> > > > >
> > > > > - Implementation detail: We estimate contextual vector embeddings of images directly from the learned segmentation network by applying Global Average Pooling (GAP) to the feature generated from the backbone feature encoder, which gives us a 2048-dimensional vector. We further use the L2 norm to normalize the vector to get an embedding on the unit hypersphere to represent the scene context embedding.
> > > > >
> > > > > - Results: The results are shown in the table below. We observe that while achieving the same performance on novel classes, the model trained with this GAP scene context embedding (without the use of an external dataset or model) even outperforms the original contextual guidance implementation on the base classes. In addition, we would like to note that while different choices of contextual guidance lead to slightly varied performance, **all of them significantly outperform the baselines**. So in practice, our method is robust to the choice of contextual guidance.
> > > > >
> > > > > | Method  | Base | Novel | HM |
> > > > > |  ----------- | -----------    | ----------- | ----------- |
> > > > > | PIFS(L2)+GAPS (w/o contextual guidance) | 48.2 +- 0.2 | 25.0 +- 0.7 | 32.9 |
> > > > > | PIFS(L2)+GAPS (w/ Places365 external model) | 48.8 +- 0.2 | 25.8 +- 0.6 | 33.7 |
> > > > > | PIFS(L2)+GAPS (w/ GAP)| 49.1 +- 0.2 | 25.8 +- 0.6 | 33.8 |
> > > > >
> > > > > We hope this experiment can address the reviewer’s concern. We will also include this result in the revision. We are happy to discuss if the reviewer has further questions.

---

> > > > > > ### Author Response · Authors · 2022-12-10
> > > > > > **Follow up to the response**
> > > > > >
> > > > > > Dear reviewer 2ZbE,
> > > > > >
> > > > > > This is a follow-up response as a friendly reminder that we have updated the response, in case you have not had the time to check the response.
> > > > > >
> > > > > > We appreciate your time and effort in reviewing and wonder if our updated responses address your concern.

---

> > > > > > > ### Comment · Reviewer_2ZbE · 2022-12-12
> > > > > > > **Follow up to the response**
> > > > > > >
> > > > > > > Thank you for your extensive answer.
> > > > > > > I am glad to see that Contextual Guidance works without external supervision and datasets.
> > > > > > >
> > > > > > > I have no further questions.

---

### Official Review · Reviewer_bVGy · 2022-10-24

**Confidence:** 5
**Correctness:** 3
**Technical Novelty And Significance:** 2
**Empirical Novelty And Significance:** 2
**Recommendation:** 5

**Clarity, Quality, Novelty And Reproducibility:**

The paper is well-structured, but it is no well-written, such as figure1 doesn't reflect the task, eq.3 mentioned in paper does not exist. The novelty of this paper is not enough, main idea is about a copy-paste technique applied to incremental few-shot segmentation. There are still much space for polishing the paper.

**Strength And Weaknesses:**

Strength:
 (1) This paper introduces copy-paste as a synthesis technique to few-shot incremental segmentation.
 (2) Experiments show clear improvement on performance with the proposed method.

Weaknesses:
(1)  My main concern is the novelty of the paper, copy-paste technique is more about a data augmentation methods, which has already been shown effective in many segmentation task, it is not novel enough to be accepted as an ICLR paper.
(2) The paper is not clear at least for me, especially figure 1 and figure, it's really hard for readers to understand the task, although the idea is clear.


**Summary Of The Paper:**

This paper proposes a few-shot incremental semantic segmentation methods via guided copy-paste synthesis. To  achieves this, authors provide three kinds of guidance, i.e., diversity-guide, context-guide, and frequency-guide. Ablation study and comparison with baseline model show clear performance improvement with copy-paste synthesis.

**Summary Of The Review:**

This paper introduces a copy-paste techniques to incremental few-shot segmentation, and designs three guidance to generate more representation training images. The novelty is a concern, because  copy-paste in segmentation is more about a data augmentation technique, which has already been proved to be effective.  Extensive experiments show clear performance improvement with such technique.

---

> ### Author Response · Authors · 2022-11-19
> **Response for Reviewer bVGy**
>
> We thank the reviewer for the valuable comments. Below we address all the concerns.
>
> ### 1. Copy-paste is not a novel technique
>
> We respectfully disagree with the reviewer that our technical approach is not novel. While the idea of copy-paste is not new in the **offline** segmentation task, our work shows that **copy-paste is less effective in the online few-shot** segmentation task and proposes a solution that is even effective when “involving only one labeled instance of the new classes” as mentioned by Reviewer 2ZbE. More specifically,
>
> - To the best of our knowledge, our work is the first to develop a novel guided copy-paste synthesis that is effective in the challenging **online few-shot** segmentation task. Notably, different from and orthogonal to the existing few-shot incremental segmentation methods that primarily focus on designing better **discriminative** models, our work shows that **data synthesis** is an effective way to deal with few-shot incremental segmentation.
>
> - We would like to further emphasize that naively applying copy-paste as an augmentation technique as described in [1] for online few-shot incremental learning is sub-optimal, due to 1) the long-tailed distribution between base classes and novel classes, and 2) the limited size of memory-replay buffer from the incremental setting.
>
> - Our work proposes three-level guidance – diversity-guide, context-guide, and frequency-guide – that enables copy-paste to overcome the aforementioned challenges. Importantly, our method is **model-agnostic** – it can be combined with a variety of few-shot incremental segmentation methods, from simple fine-tuning to more sophisticated SSUL and PIFS models, and consistently improve their performance. It is precisely the ability to be used as a plug-and-play component that makes our method simple, general, and very easy to reposeduce.
>
> - Empirically, the first row in the table below (which is the third row in the updated ablation) shows that simple application of copy-paste with all our proposed guidance techniques turned off yields sub-optimal results. We can see that our guidance techniques contribute +1.8 base IoU and +6 novel IoU to simple copy-paste, which is significant.
>
> | Method  | Base | Novel | HM |
> |  ----------- | -----------    | ----------- | ----------- |
> | PIFS(L2)+SimpleCopy  | 47.0 +- 0.2 | 19.8 +- 0.6 | 27.9 |
> | PIFS(L2)+GAPS | 48.8 +- 0.2 | 25.8 +- 0.6 | 33.8 |
>
> [1] Ghiasi, Golnaz, et al. "Simple copy-paste is a strong data augmentation method for instance segmentation." CVPR. 2021.
>
> ### 2. Paper Writing
>
> We apologize for any confusion from the writing of the paper. We have made the following amendments to the paper in the hope that it makes the writing easier to follow, and further suggestions on where improvements are needed are always welcomed.
>
> - We have updated figure 1 in the paper to use consistent terminology with other parts of the paper in the hope that it would provide clearer intuition of the task setting.
>
> - We removed the missing Eq. (3) reference to related work.

---

> > ### Comment · Area_Chair_S9Mv · 2022-11-23
> > **Following up on the rebuttal**
> >
> > Dear reviewer,
> >
> > like with the reviewer oYb5, the authors tried to address your comments, most notably on the account of novelty, clarifying that the focus on the few-shot incremental semantic segmentation. Has the rebuttal in general influenced your view?
> >
> > Cheers, Your AC

---

> > > ### Author Response · Authors · 2022-12-10
> > > **Following up to the response**
> > >
> > > Dear reviewer bVGy,
> > >
> > > This is a follow-up response as a friendly reminder that we have updated the response, in case you have not had the time to check the response. Do our updated responses address your concern?

---

### Official Review · Reviewer_oYb5 · 2022-10-25

**Confidence:** 4
**Clarity, Quality, Novelty And Reproducibility:** Please see above for detailed comment…
**Correctness:** 3
**Technical Novelty And Significance:** 2
**Empirical Novelty And Significance:** Not applicable
**Recommendation:** 5

**Strength And Weaknesses:**

Strengths:
- The paper introduces the copy-paste data augmentation strategy to the few-shot incremental semantic segmentation, which seems to be a novel combination.
- The proposed guided sample selection method for memory construction and data synthesis improves the performance of several baseline methods.
- The paper is most clearly written and easy to follow.

Weaknesses:
- While the proposed data augmentation strategy helps few-shot learning, its technical novelty in the incremental semantic segmentation (ISS) is rather limited, as it is an add-on module and has to rely on the base ISS methods.
- The construction of the memory-replay buffer has several limitations and will affect the quality of the data augmentation in ISS:
1) Only the base stage has fully-annotated images that can be used for the target images in the copy-paste operation, and all the images from the incremental stages are partially labeled, which are unusable for that purpose. As a result, when the incremental learning evolves, the memory has to add samples of new classes and remove some base-stage images due to the capacity limit. Eventually, the memory buffer would be less useful.
2) The diversity-guided selection procedure in Sec 3.2 considers each class separately in sampling. However, for semantic segmentation, it is common that each image has regions of multiple classes. It is unclear how such a method can balance different classes. This is also a problem for the class-frequency-guided selection, as each image can have multiple classes. Another issue is that this method did not take into account the region sizes as it only considers the image-level sampling factor.
- There are several concerns on the experimental evaluation, which make the conclusion less convincing:
1) Both benchmarks are mainly used in the few-shot segmentation literature, which are a bit small in terms of data set size and number of classes. Similar to SSUL and other ISS works, the method should be evaluated on more diverse ISS task settings and larger dataset, such as ADE.
2) For the few-shot side, while this work adopts a different evaluation protocol (as described in Sec. 4.2), it would be more informative if it also uses the protocol in Cermelli et al 2021 for a fair comparison. Moreover, unlike the standard ISS, the training process of the few-shot setting typically has a large variance due to different training samples annotated. Thus it should report both the mean and standard deviation in the final results.
3) The gaps in the ablation study are relatively small. It would more convincing if the standard deviation can be provided. It is unclear how significant such improvements are in the few-shot learning setting.

**Summary Of The Paper:**

The paper presents a copy-paste data augmentation strategy for the few-shot incremental semantic segmentation task.  It starts with a base learning stage to train an initial segmentation network with fully-annotated images. In the subsequent incremental learning stages, the proposed method develops a guided sample selection strategy to build a memory-replay buffer and to generate fully-labeled training examples with the copy-paste technique from the buffer and a small set of images with novel classes annotated. The guided sample selection strategy considers three factors: diversity in difficulty levels, context similarity, and class frequency balance. The authors integrate their strategy with several incremental segmentation pipelines and evaluate them on two few-shot segmentation benchmarks, PASCAL-5i and COCO-20i.

**Summary Of The Review:**

While the proposed copy-and-paste strategy seems interesting for the few-shot ISS task, there are several concerns on this work, including the novelty of the method for ISS, the effectiveness of the memory for data synthesis in ISS and the insufficient experimental evaluation. As a result, my preliminary rating leans toward the negative side.

---

> ### Author Response · Authors · 2022-11-19
> **Response for Reviewer oYb5 (part 1/3)**
>
> We thank the reviewer for the valuable comments. Below we address all the concerns.
>
> ### 1. The technical novelty with respect to conventional incremental semantic segmentation (ISS)
>
> We respectfully disagree with the reviewer that the technical novelty of our method with respect to standard incremental semantic segmentation (ISS) is a weakness, for the following reasons:
>
> - First, we would like to clarify that **our paper focuses on the few-shot ISS setting rather than the conventional ISS**. As mentioned by **Reviewer 2ZbE**, “the paper investigates a novel and understudied scenario that is more realistic than earlier benchmarks.” Indeed, the few-shot ISS is more realistic and much more challenging than conventional ISS, because of **1) scarcity of data and 2) long-tailed distribution of base classes and novel classes**.
>
> - In our experiment, we have shown that the performance of the widely-used ISS methods dramatically drops in this challenging few-shot setting. For example, the novel IoU of SSUL, a state-of-the-art standard ISS method, drops from 43.4 (in the ISS setting) to 21.7 when it is evaluated under the 5-shot setting. Similar findings have also been discovered in previous works on few-shot incremental object classification [1] and few-shot incremental object detection [2]. These works demonstrated that it is important to take special care of method design for the challenging few-shot incremental learning setting, and call for new methods that are tailored to few-shot ISS.
>
> - Second, we respectfully disagree with the reviewer that our method “is an add-on module and has to rely on the base ISS methods.” Different from and orthogonal to the existing ISS methods that primarily focus on designing better **discriminative** models, our work shows that **data synthesis** is an effective way to deal with few-shot ISS. It is precisely the ability to be used as a plug-and-play component that makes our method simple, general, and very easy to reposeduce. Moreover, our data synthesizer does no reply on a particular base ISS method; instead, our experiment (Table 1) shows that our GAPS method is **model-agnostic** – GAPS can be combined with a variety of base ISS methods, from simple fine-tuning to more sophisticated SSUL and PIFS models, and consistently improve their performance in the few-shot ISS scenarios.
>
> [1] Tao, Xiaoyu, et al. "Few-shot class-incremental learning." CVPR. 2020.
>
> [2] Perez-Rua, Juan-Manuel, et al. "Incremental few-shot object detection." CVPR. 2020.

---

> > ### Author Response · Authors · 2022-11-19
> > **Response for Reviewer oYb5 (part 2/3)**
> >
> > ### 2. Construction of memory-replay buffer: Partially-annotated images are not used as background for copy-pasting
> >
> > We agree with the reviewer that for the construction of the memory-replay buffer, there is a trade-off between the number of fully-annotated images from the base stage and the number of partially-annotated images from the incremental stages; and only the former is used in the copy-paste operation. However, this issue has minimal impact on our method in practice, for the following two reasons:
> >
> > - Our performance is **not sensitive** to this trade-off, as we do not require a large number of fully-annotated images from the base stage. To validate this point, we conducted some additional experiments on the COCO-20-1 dataset under the 5-shot setting, where we vary the number of fully-annotated images in the memory-replay buffer to investigate the performance of our method. As can be seen from the table below and **Figure 4 in the appendix**, even with as few as 100 images, GAPS still consistently boosts the performance of the underlying ISS method.
> >
> > | Memory Size | Base | Novel | HM |
> > |  ----------- | -----------    | ----------- | ----------- |
> > | 100 | 47.0 +- 0.3 | 22.9 +- 0.7 | 30.8 |
> > | 200 | 47.4 +- 0.3 | 23.5 +- 0.7 | 31.4 |
> > | 300 | 47.6 +- 0.3 | 24.3 +- 0.7 | 32.2 |
> > | 400 | 48.2 +- 0.2 | 23.6 +- 0.8 | 31.7 |
> > | 500 | 48.9 +- 0.2 | 25.3 +- 0.9 | 33.3 |
> >
> > - In addition, we have revised the proposed method in the paper accordingly. After the revision, a minimum threshold (80%) of fully-annotated images is enforced in the memory-replay buffer for copy-paste. Note that such revision does not affect the numbers reported in the paper because existing incremental learning benchmarks do not encounter such an asymptotic case.
> >
> > - Finally, from a perspective of practical applications, this issue is relatively insignificant, because 1) storing partially-annotated images can be done by storing annotated instances only, which is much more storage-efficient than storing full images, as novel object instances take up only a small portion of the image; 2) due to the advance in storage technology, in practical application, constructing a relatively large memory-reply buffer is feasible and in fact widely-adopted in practice. Some recent work on continual learning even uses >= 1,000 samples for the memory-reply buffer [3, 4, 5, 6].
> >
> > [3] Bang, Jihwan, et al. "Rainbow memory: Continual learning with a memory of diverse samples." CVPR. 2021.
> >
> > [4] Mai, Zheda, et al. "Supervised contrastive replay: Revisiting the nearest class mean classifier in online class-incremental continual learning." CVPR. 2021.
> >
> > [5] De Lange, Matthias, and Tinne Tuytelaars. "Continual prototype evolution: Learning online from non-stationary data streams." ICCV. 2021.
> >
> > [6] Koh, Hyunseo, et al. "Online Continual Learning on Class Incremental Blurry Task Configuration with Anytime Inference." ICLR. 2021.

---

> > > ### Author Response · Authors · 2022-11-19
> > > **Response for Reviewer oYb5 (part 3/3)**
> > >
> > > ### 3. Construction of memory-replay buffer: Proposed diversity-guided selection technique does not consider multiple classes in an image, and does not consider region sizes.
> > >
> > > We thank the reviewer for the comment. As suggested by the reviewer, we implemented a few more baselines that consider multiple classes and consider region sizes in the selection procedure. The comparisons show that our original diversity-guided selection still works the best.
> > >
> > > - To consider multiple classes in images, we design two baseline methods. RFS (Repeat Factor Sampling) is done by using the weighted sampling technique described in LVIS [7], which considers multiple classes in samples by computing category-level re-sampling factor and creates image-level re-sampling factor based on the maximum category-level factor in the image. RFS assigns higher weights to classes with less examples and has been shown to work well in long-tailed settings. The other baseline, ‘GreedyClass,’ is performed by greedily picking samples in every base class with the most number of classes. For instance, given a sample A with 3 unique classes in the image and a sample B with 5 unique classes in the image, sample B will always be preferred over A.
> > >
> > > - To consider the region sizes in the image, we design a baseline method, ‘BalancedRegion,’ which uses a similar algorithm as our proposed diversity sampling method, but we replace the difficulty estimation with the number of pixels which the selected class occupies in images.
> > >
> > > - The table below summarizes the comparison. We have also included more details and discussions in Section A.2 in the revised Appendix. In particular, the ‘GreedyClass’ method improves over the random baseline method due to the weighted sampling samples with more classes; and the ‘BalancedRegion’ method considers the sizes of objects from corresponding classes, which leads to increased geometric variation and more balanced pixel-level supervision. However, both of the suggested methods fail to outperform our proposed difficulty-based diversity method. In summary, difficulty metrics is the best indicator of sample diversity among these three methods. It may be possible to design an even better method that considers all of the above factors, which we leave as interesting future work.
> > >
> > > | Method  | Base | Novel | HM |
> > > |  ----------- | -----------    | ----------- | ----------- |
> > > | RFS  | 47.6 +- 0.3 | 25.2 +- 0.7 | 33.0 |
> > > | GreedyClass | 48.7 +- 0.2 | 25.0 +- 0.7 | 33.0 |
> > > | BalancedRegion | 48.6 +- 0.2 | 24.9 +- 0.8 | 32.9 |
> > > | DiverseSampling (Ours) | 48.9 +- 0.2 | 25.3 +- 0.9 | 33.3 |
> > >
> > > [7] Gupta, Agrim, Piotr Dollar, and Ross Girshick. "LVIS: A dataset for large vocabulary instance segmentation." CVPR. 2019.
> > >
> > > ### 4. Evaluation on ADE20K
> > >
> > > Per the reviewer’s request, to validate our approach on the ADE dataset, we ran an experiment where we follow the ADE100-5 (11 steps) setup discussed in increment segmentation literature (e.g., SSUL) and modify it as 5-shot learning. We tested PIFS(L2) and PIFS(L2)+GAPS on this setting. Again, from the table below we observe a steady increase in both the base IoU and novel IoU in this challenging task setting on methods augmented with GAPS. This further validates the effectiveness of our approach.
> > >
> > > | Method  | Base | Novel | HM |
> > > |  ----------- | -----------    | ----------- | ----------- |
> > > | PIFS(L2)  | 33.8 +- 0.9 | 3.5 +- 1.5 | 6.3 |
> > > | PIFS(L2)+GAPS | 36.2 +- 0.6 | 5.9 +- 1.4 | 10.1  |
> > >
> > > ### 5. Should adopt the same evaluation protocol as PIFS [Cermelli et al. 2021]
> > >
> > > We apologize for the confusion. For the evaluation protocol, the table did adopt the same protocol as PIFS, which averages the results of base and novel IoU after every incremental learning step is completed. To avoid possible confusion, we have revised Sec. 4.2 to clarify our evaluation protocol.
> > >
> > > ### 6. Multiple runs and standard deviations for ablation study
> > >
> > > We thank the reviewer for the suggestion to help better understand the variance due to the varying quality of supplied samples. In fact, as described in Sec 4.2, numbers reported in the ablation study have considered the variance due to varying quality of supplied samples by averaging results over multiple runs. **For clarification, we have revised the main ablation study table (Table 2) in the paper PDF to include 95% confidence intervals over 20 trials for the ablation study following existing few-shot learning literature [8]**. It is evident that all proposed guidance techniques raise the performance in a statistically significant manner: 1) frequency guidance (F-guide) contributes mostly to novel class performance; 2) diversity guidance (D-guide) diversifies memory-replay buffer and increases base class performance; 3) contextual guidance helps performance on both base and novel classes.
> > >
> > > [8] Chen, Wei-Yu, et al. "A Closer Look at Few-shot Classification." International Conference on Learning Representations. 2019.

---

> > > > ### Comment · Area_Chair_S9Mv · 2022-11-23
> > > > **Following up on the rebuttal**
> > > >
> > > > Dear reviewer,
> > > >
> > > > the authors tried to address your comments, most notably on the account of novelty, clarifying that the focus on the few-shot incremental semantic segmentation. Has the rebuttal in general influenced your view?
> > > >
> > > > Cheers,
> > > > Your AC

---

> > > > > ### Comment · Reviewer_oYb5 · 2022-11-25
> > > > > **Post-rebuttal comments**
> > > > >
> > > > > Thanks the authors for the detailed responses, which partially addressed the initial concerns, including Q3, Q4, Q5 and Q6. However, as discussed below, I am not fully convinced by the replies to Q1 (novelty) and Q2 (memory), which seems to be the major weaknesses of this work.
> > > > > 1. Novelty: My original comment is not about the novelty of this problem setting, but on the technical contribution of the proposed method. As noted by other reviewers, the copy-and-paste method for data augmentation is not new. And this work heavily relies on the existing incremental SS strategies, which is confirmed by the authors' rebuttal: the proposed technique has to be combined with some existing ISS method for evaluation.
> > > > > 2. Memory construction: The revised memory strategy partially addressed my concern on the memory. However, given  limited fully-annotated base-stage data, the proposed data augmentation is prone to overfitting after long incremental steps.

---

> > > > > > ### Author Response · Authors · 2022-11-28
> > > > > > **Response to follow-up questions on Q1 (novelty) and Q2 (memory) (part 1/2)**
> > > > > >
> > > > > > We thank the reviewer for the response, and we are happy to see that the concerns for Q3-Q6 have been addressed. Below we would like to further clarify Q1 (novelty) and Q2 (memory), and hope can address the reviewer’s remaining concerns.
> > > > > >
> > > > > > ## Q1: novelty
> > > > > >
> > > > > > We thank the reviewer for clarifying the question. In our previous response, we clarified the technical contribution of the proposed method (the third bullet point). Regarding the reviewer’s two concerns:
> > > > > >
> > > > > > ### 1.1 The copy-and-paste method for data augmentation is not new
> > > > > >
> > > > > > While the idea of copy-paste is not new in the **offline** segmentation task, our work shows that **conventional copy-paste fails in the more challenging online few-shot** segmentation task, and we propose a solution that is even effective when “involving only one labeled instance of the new classes” as mentioned by Reviewer 2ZbE. Our key contribution in this work is to introduce the critical different-level guidance techniques rather than naively applying copy-paste. To the best of our knowledge, our work is the first to develop a novel guided copy-paste synthesis that is effective in the online few-shot segmentation task. Specifically,
> > > > > >
> > > > > > - We would like to emphasize that naively applying copy-paste as an augmentation technique as described in [1] for online few-shot incremental learning leads to poor results, due to 1) the long-tailed distribution between base classes and novel classes, and 2) the limited size of memory-replay buffer from the incremental setting – that is, the discrepancy between offline and online few-shot settings.
> > > > > >
> > > > > > - Our work proposes three-level guidance – diversity-guide, context-guide, and frequency-guide – that enables copy-paste to overcome the aforementioned challenges.
> > > > > >
> > > > > > - Empirically, the second row in the table below (which is the third row in the updated ablation in Table 2) shows that using conventional naive copy-paste with all our proposed guidance techniques turned off yields poor results. We can see that **our guidance techniques contribute +1.8 base IoU and +6 novel IoU to conventional copy-paste, which is significant**.
> > > > > >
> > > > > > | Method  | Base | Novel | HM |
> > > > > > |  ----------- | -----------    | ----------- | ----------- |
> > > > > > | PIFS(L2) (w/o CopyPaste) | 46.2 +- 0.3 | 20.2 +- 0.7 | 28.1 |
> > > > > > | PIFS(L2)+NaiveCopyPaste [1]  | 47.0 +- 0.2 | 19.8 +- 0.6 | 27.9 |
> > > > > > | PIFS(L2)+GAPS (Ours) | 48.8 +- 0.2 | 25.8 +- 0.6 | 33.8 |
> > > > > >
> > > > > > [1] Ghiasi, Golnaz, et al. "Simple copy-paste is a strong data augmentation method for instance segmentation." CVPR. 2021.
> > > > > >
> > > > > > ### 1.2 The proposed technique has to be combined with some existing ISS methods for evaluation.
> > > > > >
> > > > > > We respectfully disagree with the reviewer that 1) “our method has to be combined with the existing ISS methods”; and 2) “our method **can be** combined with the existing ISS methods” indicates that “our method **heavily relies on** the existing ISS methods,” for the following reasons:
> > > > > >
> > > > > > - Our GAPS can be used **with simple fine-tuning without any existing ISS methods**. In fact, as shown in Table 1, our FINETUNE+GAPS already leads to a performance that is significantly better than existing ISS methods like SSUL and PIFS. For easy reference, we list the performance on the COCO-20i dataset under the 5-shot setting in the table below.
> > > > > >
> > > > > > | Method  | Base | Novel | HM |
> > > > > > |  - | - | - | - |
> > > > > > | FT + GAPS | 46.4 | 24.9 | 32.4 |
> > > > > > | SSUL | 51.6 | 15.0 | 23.2 |
> > > > > > | PIFS(L2) | 46.2 | 20.2 | 28.1 |
> > > > > >
> > > > > > - As a general data synthesis method, our GAPS can be also combined with **a variety of** ISS methods to improve their performance consistently. However, our GAPS **does not rely on a particular** ISS method or its design.
> > > > > >
> > > > > > - We would like to emphasize that the aim of this paper is not to propose a new “discriminative” ISS method. Instead, we see data synthesis as an effective way to deal with few-shot ISS which is significantly under-explored in the existing literature. Therefore, our work is orthogonal to the existing ISS methods that primarily focus on designing better discriminative models.

---

> > > > > > > ### Author Response · Authors · 2022-11-28
> > > > > > > **Response to follow-up questions on Q1 (novelty) and Q2 (memory) (part 2/2)**
> > > > > > >
> > > > > > > ## Q2: Memory
> > > > > > >
> > > > > > > We thank the reviewer for the follow-up comment. We would like to further clarify from the following aspects:
> > > > > > >
> > > > > > > - First, as noted by the reviewer, our revised strategy now maintains a minimum portion of fully-annotated base-stage data, which technically avoids the scenario where the memory falls short of fully-annotated base data for copy-pasting over the long run, and makes our method comparable to general memory-replay-based methods in continual learning.
> > > > > > >
> > > > > > > - Second, overfitting to a small memory-replay buffer is a general and open question in continual learning, which is not specific to our work.
> > > > > > >
> > > > > > > - In fact, **our GAPS is less prone to overfitting than the baseline after long incremental steps**. This has been validated in Section A.5 and Figure 3 in the original Appendix: we investigated the performance of our method in a more long-term setting, where we apply a **20-step incremental learning** process on the COCO dataset. Figure 3 includes the full results, and here we show the results after the 20 steps in Figure 3 as a table below for easy reference. We can see from Figure 3 in the Appendix that the slope of decreasing base IoU is considerably slower with the memory-replay method in GAPS – the performance gain of our GAPS over the beeline is even more pronounced after the long-term continual learning steps.
> > > > > > >
> > > > > > > | Method | Base | Novel | HM |
> > > > > > > |  - | - | - | - |
> > > > > > > | PIFS(L2) | 25.2 | 2.8 | 5.0 |
> > > > > > > | PIFS(L2)+GAPS | 43.5 | 21.5 | 28.8 |

---

> > > > > > > > ### Author Response · Authors · 2022-12-10
> > > > > > > > **Following up on the response**
> > > > > > > >
> > > > > > > > Dear reviewer oYb5,
> > > > > > > >
> > > > > > > > This is a follow-up response as a friendly reminder that we have updated the response, in case you have not had the time to check the response.
> > > > > > > >
> > > > > > > > We appreciate your time and effort in reviewing and wonder if our updated responses address your concern.

---

### Author Response · Authors · 2022-11-19
**General response**

We sincerely thank all the reviewers for their constructive and insightful comments. In addition to editing the manuscript to reflect changes suggested by the reviewers, we have also conducted additional experiments and listed the results under individual responses and the appendix of the manuscript to address the concerns raised by the reviewers.

Major changes to the main text of the manuscript are highlighted in yellow in the updated PDF and summarized here:

- We revised the terminology and caption of Figure 1 as suggested by Reviewer bVGy.

- We revised Section 3.2 to revise the proposed method to maintain a minimum proportion of fully-annotated images as suggested by Reviewer oYb5, and also removed the confusing reference to Eq. (3) in a related paper as suggested by Reviewer bVGy.

- We revised Section 4.2 to clarify that we are using the same evaluation protocol as previous work as suggested by Reviewer oYb5.

- We added 95% confidence intervals to Table 2 as suggested by Reviewer oYb5, and added two additional rows to show that the improvement of our proposed method does not solely depend on the exemplars as suggested by Reviewer 2ZbE.

- Experiments suggested by reviewers are added in the appendix: major revision to Sec A.2 to reflect new exemplar construction experiments. New sections A.6 A.7 A.8 A.9 are added to reflect the requested new experiments

Other minor changes (e.g., correcting typos; grammatical fixes) are not included here.

We hope that our responses have addressed the reviewers’ concerns, and we are looking forward to further discussions and suggestions.

---

### Decision · Program_Chairs · 2023-01-20

**Decision:**

Reject

**Justification For Why Not Higher Score:**

The paper is not clear in its empirical findings. Several more experiments, variants, and baselines were implemented during the rebuttal, such that a resubmission is warranted.

**Justification For Why Not Lower Score:**

The problem motivation is interesting.

**Metareview: Summary, Strengths And Weaknesses:**

The paper proposes guided copy-paste synthesis for few-shot incremental semantic segmentation. The copy-paste method is a data augmentation method, where image segments are transferred from one image to other images. As the authors argue, this does not work that well in an online setting, and for that they proposed to guide the process. While the method seems promising, the manuscript seems to have a few unclarities that prevent it from being published.

For one, I am not entirely convinced about how to sampling is guided by context so that the copy-paste is makes sense. In the manuscript the Places365 is used to obtain a scene embedding, which is quite limiting, for instance context is not just about the scenery. A similar critique was put forward also by reviewer 2ZbE. Authors did an experiment with using as embedding a vector by global pooling from the neural network itself, and they also did an experiment where they skipped contextual guidance. Using the global pooling vector performed on par, and the one with no contextual guidance did a bit worse. In either case, it casts doubt on what is exactly the contribution of contextual guidance in the copy-pasting, which is one of the main contributions of the work: is contextual guidance needed, is it not needed, when is it needed?

Besides this point, there were a few other points to address. Can the model handle well settings with multiple classes present? The authors came up at rebuttal time with a few baseline variants, that seem to underperform compared to the proposed method. However, it is a question whether one can easily judge the significance of the results when major changes are proposed at the rebuttal time. What is more, in this experiment as well as other experiments there seem to be lots of hyperparameters involved, and it must be crystal clear how these parameters influence the results. Certainly, picking '80%' arbitrarily is not good enough, as in a different setting this hyperparameter might not work, meaning poor generalization (and the whole point of continual learning is generalization).

All in all, the paper in its current form is not crisp and clean enough regarding its empirical findings, and it must be improved, with clear experiments, clear ablation studies, and clear conclusions (clear does not mean many necessarily).